# Unsupervised Object-Based Transition Models for 3D Partially Observable Environments

**Antonia Creswell**
DeepMind
London
tonicreswell@deepmind.com

**Rishabh Kabra**
DeepMind
London
rkabra@deepmind.com

**Christopher Burgess**
DeepMind
London
cpburgess@deepmind.com

**Murray Shanahan**
DeepMind
London
mshanahan@deepmind.com

## Abstract

We present a slot-wise, object-based transition model that decomposes a scene into objects, aligns them (with respect to a slot-wise object memory) to maintain a consistent order across time, and predicts how those objects evolve over successive frames. The model is trained end-to-end without supervision using transition losses at the level of the object-structured representation rather than pixels. Thanks to the introduction of our novel alignment module, the model deals properly with two issues that are not handled satisfactorily by other transition models, namely object persistence and object identity. We show that the combination of an object-level loss and correct object alignment over time enables the model to outperform a state-of-the-art baseline, and allows it to deal well with object occlusion and re-appearance in partially observable environments.

## 1 Introduction

In spite of their well-documented ability to learn complex tasks, today's deep reinforcement learning agents are still far from matching humans at out-of-distribution generalisation or few-shot transfer [6, 21, 23]. Two architectural features commonly proposed to remedy this are (1) transition models that enable the agent to internally explore paths through state space that it has never experienced [29, 25, 12], and (2) compositionally structured representations that enable the agent to represent meaningful states that it has never encountered [7]. These two features are not exclusive; transition models that operate on compositionally structured representations are a potent combination, and these are the subject of the present paper. Specifically, our focus is on transition models that operate at the level of *objects*, which are the most obvious candidates for the structural elements of representations likely to be useful for artificial agents inhabiting 3D worlds such as our own [28].

While there has been progress in object-based transition models [34, 32, 35, 20], current models do not deal satisfactorily with *object persistence* (the concept that objects typically continue to exist when they are no longer perceptible [24]) or with *object identity* (the concept that a token object at one time-step is the same token object at a later time-step [1]). As we show, transition models that neglect object persistence tend to perform badly in complex, partially observable environments, while models that neglect object identity are unable to integrate information about a single object (and its interactions) over time in a way that generalises to future time-steps. By proposing and incorporating a novel module for aligning objects across time using a slot-based memory, our model handles both these concepts, and exhibits better performance as a result.

35th Conference on Neural Information Processing Systems (NeurIPS 2021).

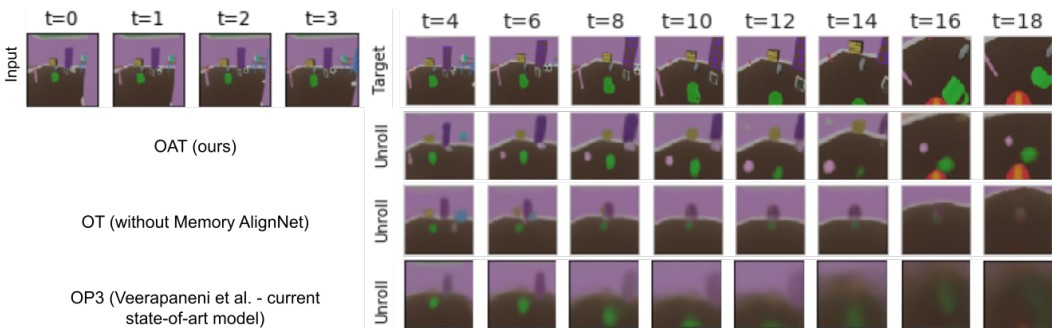

Figure 1: **Comparing our model to baselines.** Each model was trained with four input steps and to unroll for six steps, here we unroll for 15 steps. Our model, OAT, performs significantly better than OT (our model without the AlignNet) and current state-of-the-art model, OP3 [32]. See Figure 17 in the Appendix for additional OP3 roll-outs.

An important feature of our transition model is that it makes predictions and computes losses in a representation space that is divided into objects. This contrasts with existing models that make predictions in an unstructured representation space [13, 12]. Making predictions and computing losses in an object-structured representation space facilitates learning, not only because the representation space is lower dimensional than pixel space, but also because the model can exploit the fact that dynamics tend to apply to objects as a whole, which simplifies learning. However, to compute prediction losses directly over distinct object representations (rather than first mapping predictions back to pixels [34, 32]), the objects in a predicted representation must be matched with those in the target representation, which again requires a proper treatment of object persistence and identity. The result is a model that can roll-out accurate predictions for significantly more steps than seen during training, outperforming state-of-the-art for comparable models.

To achieve this, our model, Objects-Align-Transition (OAT), combines (1) a *scene decomposition and representation module*, MONet [2], that transforms a raw image into a slot-wise object-based representation, (2) an novel *alignment module* which, with the aid of a slot-wise memory, ensures that each object is represented in the same slot across time, even if it has temporarily disappeared from view, and (3) a novel slot-wise *transition model* that operates on the object representations to predict future states. All

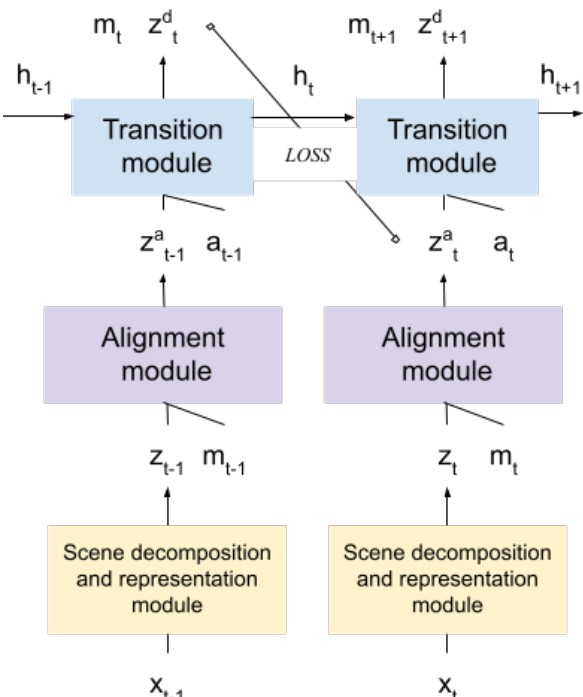

Figure 2: **Encoding steps.** The *scene decomposition and representation module* extracts $K$ object representations, $z_t \in \Re^{K \times F}$, from images, $x_t \in \Re^{W \times H \times 3}$. These are aligned using the *alignment module* to obtain $z_t^a \in \Re^{M \times F}$. Aligned objects are fed to the slot-wise *transition model* with the action, $a_t$, and hidden state, $h_t(= h_t^a) \in \Re^{M \times H}$, to predict the object representations at the next time-step, $z_t^d \in \Re^{M \times F}$, as well as the updated memory, $m_{t+1} \in \Re^{M \times F}$. We also depict the transition model loss, $l_{\text{Transition model}}$, computed between $z_t^d$ and $z_t^a$ (shown as LOSS in the figure).

three components are differentiable, and the whole model is trained end-to-end without supervision. We evaluate the model on two sequential image datasets. The first is collected from a pre-trained agent moving in a simulated 3D environment, where we show that our model outperforms the current state-of-the-art object-based transition model (OP3 [32]). The second dataset is collected from a real

robot arm interacting with various physical objects, where we demonstrate accurate roll-outs over significantly longer periods than used in training. An ablation study shows that the model's success depends on the combination of correct object alignment across time and the use of a transition loss over object-level representations instead of over pixels.

## 2 Our Model: Objects-Align-Transition

Our model, Objects-Align-Transition (OAT), combines a scene decomposition and representation module, in this case MONet, with our slot-wise transition module, via our alignment module, which ensures that each slot in the transition model receives objects corresponding to the same token object (or identity) across time. The whole model is trained end-to-end. We now provide details of each of these modules (see Figure 2).

### 2.1 Scene Decomposition and Representation Module

The input to OAT is a sequence of RGB images, $x \in [0,1]^{T \times W \times H \times 3}$, with $T$ time-steps, width, $W$, and height, $H$. We leave out the batch dimensions for simplicity. Each image in the sequence, $x_t$, is passed through a scene decomposition and representation module[1], in this case MONet, to obtain a slot-wise object representation, $[z_{t,k}]_{1:K}$, occupying $K$ slots, where $z_{t,k} \in \Re^F$ is an object representation vector with $F$ features.

MONet consists of an attention module (a U-net [27]) — which predicts object segmentation masks, $\mu_{t,k} \in [0,1]^{T \times W \times H \times 1}$, for each slot — and a slot-wise VAE. The encoder of the slot-wise VAE is fed the predicted object segmentation masks and the input image, $x_t$, and outputs object representations, $z_t \in \Re^{K \times F}$. The decoder reconstructs the masks, $\tilde{\mu}_{k,t}$, and each object's pixels, $\tilde{x}_{t,k}$. MONet's computations can be summarised by the following equations: $\mu_{k,t} = \text{Attention\_Network}(x_t)$, $z_{k,t} = \text{MONet\_encoder}(x_t, \mu_{k,t})$ and $\tilde{\mu}_{k,t}, \tilde{x}_{t,k} = \text{MONet\_decoder}(z_k)$.

It may be tempting to feed the slot-wise object representations, $[z_{t,k}]_{k=1:K}$, at time, $t$, directly to a slot-wise transition model. However, MONet's representations [2] are *not stable* across time, meaning that a specific object may appear in different slots at different times [35] (see Figure 3). This leads to two major problems when training slot-wise transition models on slot-wise object-based representations. Firstly, it is difficult to compute losses between predicted and target object slots. Computing a slot-wise *object-level loss* requires us to know how the previous object representations (and thus the predicted object representations) correspond with the target object representations. Secondly, if object representations do not occupy consistent slots it makes it harder to integrate information about a single object (and its interactions with other objects) across time [35], and makes it harder to predict the reappearance of that object (as show in Section 4.2). To address the issue of slot stability we introduce Memory AlignNet, an alignment module.

### 2.2 Alignment Module

Our alignment module, Memory AlignNet, must play two key roles. The first is to align objects across time, enabling us to compute slot-wise object-level losses for training. The second is to learn a slot-wise memory that encodes the history of each slot across time, allowing our transition model to operate effectively in partially observable environments by keeping track of objects as they go in and out of view. This is especially important for embodied agents that take actions in 3D environments, where the agent's looking around frequently causes objects to move in and out of its field-of-view.

We propose Memory AlignNet, which takes a slot-wise memory, $m_t \in \Re^{M \times F}$, with $M \geq K$ slots and the (stacked) output of the scene decomposition and representation module, $z_t \in \Re^{K \times F}$, returning the aligned object representations, $z_t^a \in \Re^{M \times F}$. To perform alignment, the Memory AlignNet predicts an adjacency matrix, $A_t \in \Re^{M \times K}$, that aligns objects, $z_t$, with the memory, $m_t$. This adjacency matrix allows us to compute a *soft* alignment $z^{a,soft} = A_t z_t$ or a hard alignment, $z^a \triangleq z^{a,hard} = \text{Hungarian}(A_t) z_t$. We will refer to $z_t^{a,hard}$ as $z_t^a$ throughout the rest of the paper. Hungarian$(\cdot)$ denotes the application of the Hungarian algorithm, a non-differentiable algorithm which computes a permutation matrix given an adjacency matrix. The soft version of the alignment is

---

[1]We tried some variants where $z_t$ depended on $z_{t-1}$. However these did not yield significantly beneficial results.

used to train the Memory AlignNet while the hard version of the alignment is passed to the slot-wise transition model, which has the same number of slots, $M$, as the memory. Further details can be found in Section A.2 of the Appendix.

The hard-aligned objects may also be used to update the memory using a recurrent slot-wise model. For simplicity we use our transition model, $\mathcal{T}_\theta(\cdot)$, to predict deltas for both the object representations, $z_t^a$, and the memory, $m_t$. We will detail this in the next section.

## 2.3  Slot-Wise Transition Module

The transition model operates in both an encoding and an unrolling phase. During the encoding phase the transition model is fed aligned, observed object representations, $z_t^a$, and actions, $a_t$, to predict aligned object representations at the next time-step, $z_{t+1}^d \in \Re^{M \times F}$. During the unroll phase, the transition model is fed the predictions, $z_t^d$, and actions, $a_t$, from the current time-step to predict the object representations, $z_{t+1}^d$, at the next time-step.

More concretely in the **encoding** steps, our transition model, $\mathcal{T}_\theta(\cdot)$, takes aligned object representations, $z_t^a$, and a hidden state, $h_t^a \in \Re^{M \times H}$, and predicts deltas for both the object representations, $z_t^a$, and memory, $m_t$. The output of our transition model is given by $[\Delta_t^a, \Delta_t^m], h_{t+1}^a = \mathcal{T}_\theta(z_t^a, a_t, h_t^a)$. The memory and the object representations are then updated as follows: $m_{t+1} = m_t + \Delta_t^m$ and $z_{t+1}^d = z_t^a + \Delta_t^a$. In the **unroll** steps, next step predictions are given by $z_{t+1}^d = z_t^d + \Delta_t^d$ where $[\Delta_t^d, \Delta_t^m], h_{t+1}^a = \mathcal{T}_\theta(z_t^d, a_t, h_t^a)$. By using the transition model to predict deltas, $z_t^d$ is aligned by default and does not need to be re-aligned when unrolling the model.

A key feature of our transition model is that weights are shared across object slots and can be instantiated in many different ways. One simple option is to use a *SlotLSTM*; an LSTM applied independently to each slot, sharing weights between slots. An alternative, novel instantiation which we found to work well is to first apply a transformer [33, 31] and then the SlotLSTM (see Figure 10 and Section A.3). This allows the model to capture interactions with other objects (via the transformer) and integrate that information over time (via the SlotLSTM). Further details are in Section A.2 of the Appendix.

## 2.4  Training

When training OAT, we jointly learn the parameters of the scene decomposition and representation module, the alignment module and the transition module. Each module has its own losses that contribute to downstream gradients and updates. For example, both the transition model and the alignment module losses can influence the scene decomposition and representation module's updates. Let us define the losses for each module.

**Scene decomposition and representation module losses (MONet)**. We train MONet using standard MONet losses, $l_{\text{MONet}}$, see Equation 3 in [2]. The first term in the MONet loss is a scene reconstruction term. This is a spatial mixture loss parameterised by $\sigma_{bg}$ and $\sigma_{fg}$, which are the standard deviation used for each slot's component likelihood (for the first slot and remaining slots, respectively) that go into the mixture loss. The remaining terms are regularisation terms that 1) induce a latent information bottleneck needed for good representation learning (scaled by $\beta$, the latent KL loss weight), and 2) ensure that predicted and target masks are similar (scaled by $\gamma$, the mask KL loss weight). MONet losses are computed only in the encoding phase and are not used in the unroll phase. Additionally, MONet loss gradients do not affect the transition module or the alignment module weights.

**Alignment module losses (Memory AlignNet)**. The Memory AlignNet is trained on a reconstruction loss between the softly aligned object representations, $z_t^{a,soft}$, and the output of the transition model, $z_t^d$, for the corresponding time-step. There are also regularisation losses including an entropy loss, $\mathbb{H}(\cdot)$, on the adjacency matrix, $A_t$, which encourages values towards zero and one; and a loss that penalises columns that sum to more that one, avoiding the case where multiple objects are assigned to the same memory slot. The AlignNet loss, $l_{\text{AlignNet}}$ is defined as $l_{\text{AlignNet}} = \sum_{t=1}^{T} ||z_t^d - z_t^{a,soft}||_2^2 + \psi \mathbb{H}(A_t) + \sum_{j=1}^{M} \max(0, (\sum_{k=1}^{K} A_{t,k,j} - 1))$, where $T$ is the total number of encoding and unroll steps. We use $\psi = 0.01$ for all experiments presented in this paper.

**Slot-wise transition module losses**. The slot-wise transition model is trained with a reconstruction loss between the outputs of the transition model, $z_t^d$, and the aligned observations for the same time-step, $z_t^a$. The slot-wise transition model loss, $l_{\text{Transition model}}$ is defined as $l_{\text{Transition model}} = \sum_{t=1}^{T} ||z_t^d - z_t^a||_2^2$. Importantly, we compute a loss directly between object representations without the need for decoding them to obtain pixels.

OAT is trained end-to-end to minimise $l_{\text{MONet}} + l_{\text{AlignNet}} + \zeta l_{\text{Transition model}}$, using $\zeta = 10$ for all experiments presented in this paper.

## 3 Related Work

Our work addresses the challenging topic of learning object-based transition models in 3D partially observable environments without supervision. We identify two phenomena that are not dealt with adequately in current object-based transition models, namely object persistence and identity. As a consequence, current models are not able to perform well in **partially observable environments**, and are often trained without **semantically meaningful losses**.

Firstly, we acknowledge previous work on temporally extended scene decomposition and representation models [35, 14] (typically used for video representation learning), object-based transition models [20, 17] and action conditional object-based transition models such as OP3 [32] and C-SWM [19]. We consider Veerapaneni et al.'s OP3 [32] to be the current state-of-the-art object-based transition model since C-SWM requires privileged information about exactly which object an action was applied to, while OP3 (like our model) simply requires the action taken by the agent. OP3 alternates between refinement steps and dynamics (or prediction) steps. One drawback of OP3 is that its refinement steps require access to the observation, which means refinement cannot be applied when rolling out the model to unobserved time-steps. For this reason Veerapaneni et al. [32] only demonstrate unrolls for a limited number of steps.

### 3.1 Current Models are not Designed for Partially Observable Environments

The world that we operate in, and which we intend our agents to operate in, is partially observable. A severe limitation of existing object-based transition models [32, 19] and temporally extended object scene decomposition and representation models [35, 11, 30, 14] is their inability to cope with partially observable environments. A promising approach proposed by He et al. [14] uses an external memory for object tracking (but does not use a transition model). Their mechanism is different to that of our proposed alignment module, AlignNet, which we use to perform slot alignment in OAT. He et al. [14] train their model, TBA, for reconstruction while AlignNet incorporates dynamics and is trained using prediction, meaning that AlignNet can resolve ambiguities using dynamics[2], while TBA cannot. Additionally, TBA can only cope with static backgrounds and so it is not applicable here. Goyal et al. [8] also develop a general a model with a notion of object persistence. However they only show results in 2D environments.

### 3.2 Current Models are not Trained Using Semantically Meaningful Losses

Another problem with current approaches to learning object-based transition models is how they are trained. In most scenes, the background accounts for most of the pixels while the objects account for only a small fraction of them. So while it may be tempting to train transition and video-representation models using a pixel-level loss [34, 35, 32, 20, 17, 14], we show that it is preferable to compute losses directly between predicted and target objects (see Section 4.2 and Figure 1). Furthermore, pixel-level losses require the object representations to be decoded into an image [34, 35, 32], which is often computationally expensive.

An alternative way to find object-level losses is to compute a minimum assignment loss [4], employing the Hungarian algorithm, between predicted and target object representations. However, doing so can be problematic because you need to first compute a similarity matrix on which to apply the Hungarian algorithm. We show in Section 4.2 that computing a loss using the Hungarian algorithm (using $L_2$

---

[2]For example, when two visually similar objects collide with one another, the AlignNet can use dynamics to resolve which object is which after the collision.

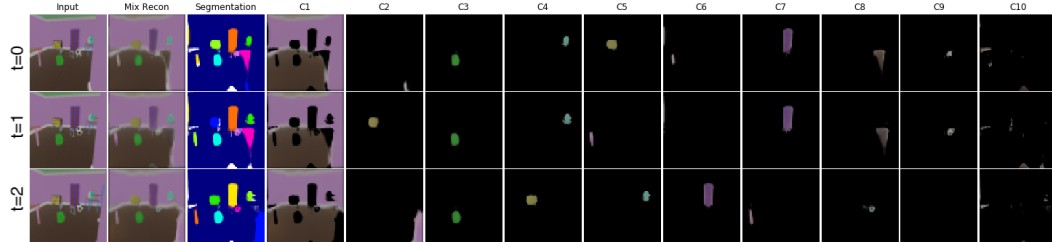

Figure 3: **Outputs of our scene decomposition and representation (MONet) module** across time. MONet predicts $K = 10$ latent object representation vectors $\{z_{t,i}\}_{i=1,...,K}$ at each time-step. Here we visualise those vectors, in columns C1 to C10, using MONet's decoder. Notice that objects switch slots across time. This makes it difficult to (a) compute losses between predictions and targets and (b) integrate information about an object across time.

loss between all object pairs as the similarity matrix) leads to poor generalisation in transition models when performing longer unrolls than those seen during training.

Furthermore, at the start of training, comparing predictions with targets to compute a loss may not be meaningful, since predictions will start off as essentially random. In our approach, we align the *inputs* with the targets, and condition the slot-wise prediction on the slot-wise history. This reinforces slot stability and allows us to accurately predict changes, $\Delta_t^a$, in each object representation.

Interestingly, Lowe et al. [22] use Deep Sets [36] to encode predicted and target object sets and use a contrastive loss between encodings. Their approach has only been demonstrated for very simple datasets. Kipf et al. (C-SWM) [19] do compute losses in their representation space. However, they avoid the object correspondence problem because they extract representations spatially, keeping the spatial ordering and using data where object movement was limited. Their approach is unlikely to scale well to partially observable environments with significant movement of objects across the field of view, especially when being trained to predict multiple steps into the future.

OP3 [32] and other models [35] attempt to induce a weak, implicit object alignment by conditioning predicted object representations on those from the previous time-step. This does not guarantee alignment (or slots with consistent identity) over time, especially in partially observable environments. Moreover, in contrast to OAT, they do not use their implicit alignment for computing losses.

Finally, we note that while there are existing models that learn to predict future states, given actions, directly from pixels [13, 12], we have focused primarily on related work that predicts the future states of objects, because we are particularly interested in developing models that may support future work towards object-based agents.

## 4 Experiments and Results

In this section we (1) demonstrate OAT's performance in the 3D Playroom environment [15, 16] and compare to the current state-of-the-art object-based transition model, OP3 [32] (Section 4.1), (2) investigate the benefit of alignment and object-level losses when training transition models (Section 4.2), and (3) apply OAT in a real world robotics environment and show it accurately predicts both the motion of the robotic arm and its physical interaction with objects (Section 4.3).

### 4.1 Objects-Align-Transition Results in Playroom

We train and test our model, OAT, on data collected by an agent taking actions in a Playroom environment [15, 16]. Each procedurally generated room in the dataset contains between 10 and 45 objects from 34 different classes in 10 different colours and three different sizes. A dataset of observation-action trajectories, $(x_t, a_t)_{t=0,1,...,20}$, is generated by an agent taking actions according to a learned policy in a procedurally generated room. We collect $100,000$ trajectories with a 7:2:1 train:validation:test split. The top row of Figure 5 shows an example trajectory.

We train OAT with four encoding steps (i.e. the model sees four observations for the first four time-steps) and six unrolling steps. MONet outputs $K$ object representations, $\{z_{t,i}\}_{i=1,...,K}$. We

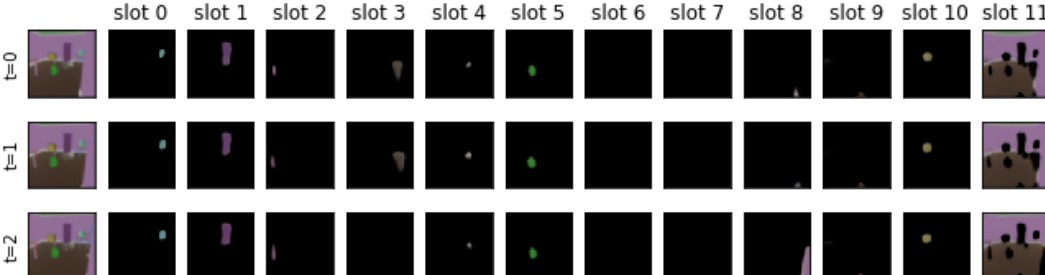

Figure 4: **Aligned inputs (first 4 time-steps) and targets (last 6 time-steps).** Our alignment module outputs object representation vectors which we visualise here using MONet's entity decoder. Most of the objects are now in consistent slots across time, making it easier to compute losses directly between object representations using a simple $L_2$ loss. Notice that while MONet outputs 10 slots, the AlignNet has 12 slots. Additional time-steps shown in Figure 11 in the Appendix.

use $K = 10$ objects, with $F = 32$ features, and a memory with $M = 12$ slots. Figure 3 visualises the MONet outputs for the first three time steps. Our model achieves good segmentation (see the Adjusted Rand Index, ARI, in Table 1 for segmentation metrics). Importantly, notice that the slots are not stable across time. For example, the purple object in slot C7 at $t = 0$ switches slots at $t = 2$. Similarly the yellow object in slot C5 at $t = 0$ switches to slot C2 at $t = 1$ and slot C4 at $t = 2$.

The object representations, $z_t \in \Re^{[K \times F]}$, output by MONet are fed to the alignment module. The outputs of the alignment module, $z_t^a \in \Re^{[M \times F]}$, are visualised in Figure 4. The outputs of the alignment module are object representations, and Figure 4 is a visualisation of these representations using MONet's decoder. Our alignment module successfully keeps objects in consistent slots across time.

The output of the alignment module is used to generate both inputs and targets for training the transition model. The transition model predicts latents, $z_t^d \in \Re^{[M \times F]}$. In Figure 5 we visualise roll-outs from our model using MONet's decoder; the reconstructed scene visualisations are generated as the mask-weighted sum of the each slot's pixels, $\sum_k \tilde{\mu}_{k,t} \tilde{x}_{t,k}$. While the model is trained to unroll for six steps, here we unroll for 15 steps. The transition model only sees the first four frames. In the top example, we see that the model is able to predict the appearance of the avatar well, and in general we notice that the model, given the agent's actions, is able to predict the position of objects well and without the representations degrading. (For comparison to baselines see Figure 1.) In some of the examples, the targets appear to have more objects than those seen in the unroll. This is because the model has only seen the first four frames and has not seen those other objects in the room, and therefore does not have enough information to predict where unseen objects will appear.

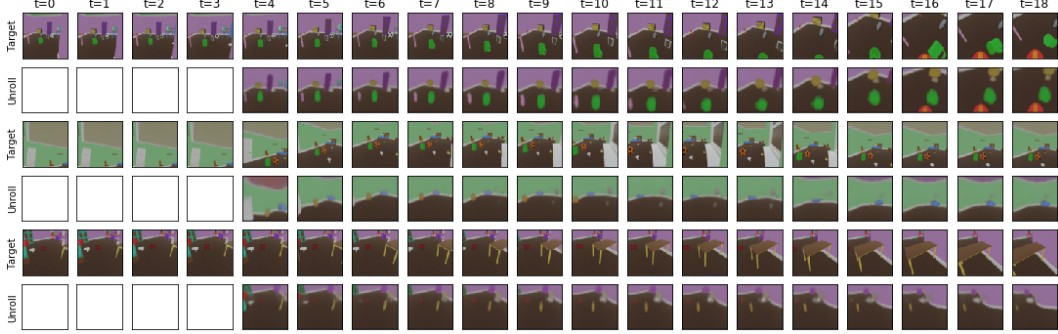

Figure 5: **Unrolling OAT for more time-steps than seen during training.** The slot-wise transition model makes predictions in latent space (one latent per object), which we visualise using MONet's decoder. The slot-wise transition model was trained with 4 inputs steps and to unroll for 6 steps. Here the model takes the first 4 steps as input and is unrolling for 15. On the top row at $t = 16$, it is impressive that the model is able predict the appearance of the avatar's base (the red circle with the yellow stripe) when looking down. (For a per-object visualisation see Figure 13 in the Appendix.)

To conclude these experiments, we compare our model to the current state-of-the-art object-based transition model, OP3 [32], and to our ablated model, OT, without the alignment module in Table 1 and Figure 1. Qualitatively, from Figure 1 we see that OAT significantly outperforms both baselines. Unrolls using OT, trained without alignment, lead to objects merging towards a grey cloud in the middle of the frame. Unrolls from OP3 lead to objects fading into the background (we explore the cause of each of these pathologies in the next Section 4.2). Results were consistent for each model across multiple runs.

To quantitatively compare models we consider three metrics: **Encoding ARI**, **Unroll Pixel Error** and **Unroll ARI**. The Encoding ARI (Adjusted Rand Index, [18, 26]) measures the accuracy of the object segmentation masks learned by the scene decomposition model. The Unroll Pixel Error and Unroll ARI evaluate the quality of the transition model's unrolls. To compute each of these we decode the latents, $z_t^d$, predicted by unrolling the transition model, to produce images, $x_t^d$, and masks, $\mu_{k,t}^d$. The Unroll Pixel Error is the mean-squared-error between the ground truth images and $x_t^d$ for the unroll steps only. The Unroll ARI is the accuracy of the decoded masks, $\mu_{k,t}^d$, compared with the ground-truth masks. For both ARI scores we exclude background pixels from the score since accurate decomposition of the objects is the main concern here. The Unroll ARI is a more meaningful evaluation of the unrolls than the Unroll Pixel Error since it is not affected by the background which often accounts for most of the pixels. Note that we only use ground-truth masks for evaluation purposes.

The results in Table 1 further demonstrate the crucial role that alignment (AlignNet) plays when performing unrolls. Without alignment the Unroll ARI is significantly lower because we are not able to compute a semantically meaningful object-level loss for training. Additionally, we see that our model significantly outperforms OP3 on all metrics (see Figure 1). In light of the results in Table 1, in the next section we will more concretely look at the role of alignment and the object-level loss (between object representations) when training transition models.

|  | Encoding ARI | Unroll Pixel Error | Unroll ARI |
|---|---|---|---|
| OAT (ours) | **0.62** | **0.0121** | **0.42** |
| OT (ours, no AlignNet) | 0.59 | 0.0143 | 0.12 |
| OP3 | 0.32 | 0.0132 | 0.33 |

Table 1: **Comparing our model OAT to baselines.** We trained three OAT and OT models (three seeds each), and report the ARI score, Unroll Pixel Error and Unroll ARI score for the model with the best Unroll ARI score. Results for OP3 were obtained similarly, except that ten models were trained (ten seeds), since more variance was observed in the OP3 results (see Appendix D for details).

## 4.2 What Matters in Object-Centric Transition Models?

In this section we demonstrate the need for both (1) **alignment**, which ensures that each slot in the transition model receives the same object consistently across time, and (2) **object-level loss**, computed between predicted and target object representations, rather than a pixel-level loss. We also compare different transition model cores and find a transformer followed by a slot-wise LSTM to be best.

In this section (Section 4.2) only, we use a MONet model that is trained using ground-truth masks instead of learning its own masks. We do this here to directly analyse the benefits of aligned object representations and object-level losses for the transition model without other confounds. Using ground-truth masks allows us to know each object's true identity across time and to directly measure the role of alignment in transition models without confounding errors from AlignNet or MONet's segmentation quality. We use a pre-trained MONet (with fixed weights) to compute object representations to keep the model similar to the full end-to-end model described in the rest of the paper.

We train slot-wise transition models under four different conditions: feeding aligned or unaligned object representations (and target) object representations to the transition model[4], and training the

---

[4]For the unaligned inputs, we shuffle the order of the ground-truth masks $\in \Re^{[K,W,H,3]}$ that are fed to MONet along the $K$ axis. For aligned inputs, we ensure that each slot contains a consistent object across time.

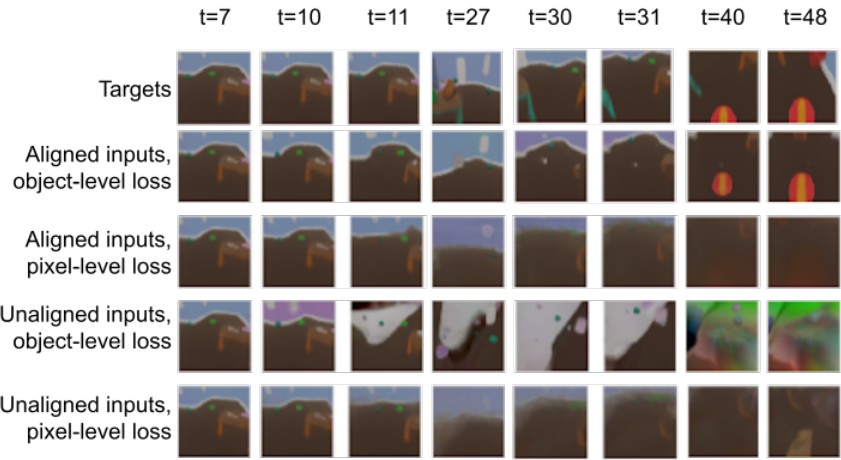

Figure 6: **Comparing slot-wise transition models trained with aligned vs. unaligned input latents and pixel-level vs. object-level loss.** These models were trained with 8 inputs steps and 12 unroll steps. Here they are being unrolled for 40 time-steps[3] The output of the transition model is a slot-wise object-based representation for each time-step. Here, we visualise the object representation vectors using MONet's decoder. Only models trained with aligned inputs were able to predict the reappearance of the chair at $t = 30$. Only the model trained with aligned inputs and object-level loss is able to predict the appearance of the avatars "base". In models trained with pixel-level loss predictions becomes very blurred.

Reporting object-level error for models trained using:

|  | Object-level loss | Pixel-level loss |
|---|---|---|
| Aligned inputs | **0.0929 ± 0.00469** | 14.4 ± 18.4 |
| Unaligned inputs | ≥ 20.3 ± 0.757 | ≥ 3312 ± 4050 |

Table 2: **The role of alignment and object-level loss when training transition models.** This table reports the mean±std latent error between predicted and ground truth latents (for 5 seeds). For models trained using unaligned input latents we compute the object-level loss using the Hungarian algorithm which is a lower bound on the actual value. Models trained with pixel-level loss can become very unstable resulting in high variance between runs. Individual seeds are shown in Figure 9.

model with a pixel-level or object-level loss. Results in Table 2 clearly demonstrate the benefits of (1) using aligned object representations for training transition models and (2) training transition models with an object-level loss. These results are critical for the future development of object-based transition models and, in particular, demonstrate the need for alignment. Note that for the experiments using the unaligned inputs we computed the object-level loss using the Hungarian algorithm which is a lower bound estimate of the true object-level loss (since the minimum assignment in $L_2$ may not be the correct assignment).

Figure 6 compares models trained under the four conditions listed above. We see that models trained using aligned inputs (and targets) are able to predict the re-appearance of objects while those without aligned latents are not (Figure 6, $t = 30$). We also see that models trained with an object-level loss and without alignment deteriorate quickly. Additionally, for models trained with a pixel-level loss we notice a "ghosting" effect where objects fade into the background across time, similar to the effects seen in the OP3 results (Figure 17). Pixel-level loss leads to this "ghosting" effect because most pixels in the observation are background pixels and so the background pixels dominate the loss.

We found these results to be consistent across multiple runs (see Figure 9 in the Appendix) and different choices for slot-wise transition model architectures. For the results shown in this section we used a transformer with a slot-wise LSTM. Figure 10, in the Appendix, compares object and pixel errors for different transition module cores. Training with object-level loss and aligned inputs, the transformer with slot-wise LSTM worked best.

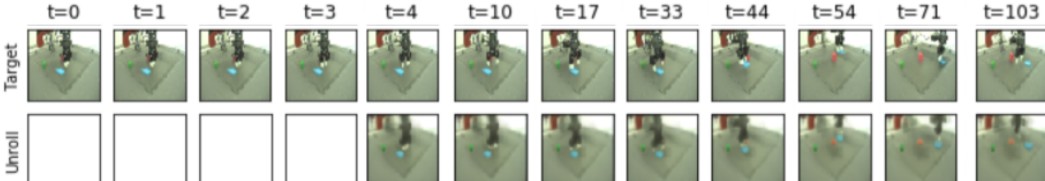

Figure 7: **Unrolling for many more steps than seen during training.** OAT is trained to take four input steps and to unroll for six time-steps. Here we demonstrate the model unrolling for 100 time-steps. Impressively, we see that the model learns both to predict the behaviour of the robot arm accurately, as well as how the arm interacts with the objects. We also see that the model is able to unroll for significantly more steps than seen during training. Note that our model makes predictions in latent space; here we are visualising those latents using MONet's decoder. Additional results in Figure 15 in the Appendix.

## 4.3 Application to Robotics

Here we demonstrate the application of OAT to a real world dataset [3] of robot trajectories. These trajectories involve a robot arm interacting with three objects of varying shapes and colours. The dataset is particularly challenging because our model must learn to predict not just the motion of the robot arm, given the arm actions, but also how the arm interacts with objects requiring some understanding of intuitive physics.

We train OAT with four input steps and to unroll for six steps. Our model achieves excellent segmentation results, shown in Figure 14 of the Appendix. Figure 7 shows impressive results obtained when unrolling the OAT model for significantly more steps than seen during training. What is more, in the third sample from the top, we see that the model correctly predicts the reappearance of the red object after it had been fully occluded at $t = 3$ for $51$ time-steps, reappearing fully at $t = 54$ in both the prediction and the target frame. Our model is able to accurately predict the reappearance of objects, even after long-term occlusion, because it explicitly captures the history of each object, endowing the model with a notion of object persistence.

## 5   Conclusion

We presented Objects-Align-Transition (OAT), an object-centric transition model that combines a scene decomposition and object representation module (MONet) with a slot-wise transition module, via an alignment module. The alignment module plays two key roles. Firstly, it ensures that the slot-wise transition model receives slot-consistent object representations across time. Secondly, it allows us to compute an object-level loss rather than a pixel-level loss which is commonly used when training transition models.

In an ablation study, we demonstrated the essential role that alignment and object-level losses play when training transition models. Additionally, we significantly outperform existing state-of-the-art object-centric transition models in a 3D partially observable environment, and we applied our model to a real-world robotics dataset, predicting many steps further into the future than seen during training.

There is room to improve our model in further work, for example by making stochastic predictions about the future, and by better modelling the uncertainty about the objects in the environment. Meanwhile, our work paves the way for future object-centric agent research, for example, enabling agents to plan over future trajectories in object representation space.

## Acknowledgments and Disclosure of Funding

We would like to thank Adam Kosiorek for feedback on early drafts of the paper as well as Arunkumar Byravan, Dushyant Rao, Yusuf Aytar and Markus Wulfmeier for their help running experiments on robotics data. This work was carried out within DeepMind, who are wholly owned by Alphabet, the parent company of Google.

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
