# A Model Details

## A.1 Objects-Align-Transition Implementation Details

### A.1.1 Training Hyperparameters

MONet and AlignNet each have their own set of hyper-parameters. OAT introduces only one additional parameter, $\zeta$, which weights the transition model loss. While we used $\zeta = 10$ for all the results in this paper, we found that the model performed similarly well for smaller values of $\zeta$, such as $\zeta = 1.0$. For MONet, we use the same hyper-parameters as those detailed in the Appendix 1.B of [2], except we use $\gamma = 0.05$ for the mask KL weight rather than $\gamma = 0.5$. Qualitatively, our model still performed well with $\gamma = 0.5$. We train OAT for 2 million steps (30 days, though the model appears to have converged after 1M steps) with an effective batch size of 32 (batch size of four spread over eight NVIDIA V100 GPUs) and a learning rate of $3 \times 10^{-4}$. We have to use a small batch size because MONet is processing a whole sequence of images and this requires more memory (for the activations) than processing a single image.

For the results on the robotics dataset we used the same hyper-parameters as above, except we use $\gamma = 0.5$. The model is trained for 1.6 million steps (16 days, however the model is close to convergence after 200k steps) using $K = 7$ object slots, $M = 8$ memory slots and a MONet feature size, $F = 32$.

## A.2 Memory AlignNet Implementation Details

In Section 2.2 of the main paper we introduced the Memory AlignNet. Here we provide further implementation details and additional details in Figure 8.

The memory, $m_t$, is initialised with zeros. At each time-step the memory, $m_t$ and unaligned latents, $z_t$, from MONet are fed to a transformer to produce an adjacency matrix, $A_t$. We append slot indices to the memory, $1, ..., M$ and set indices, $0$ to the memories and $1$ to the unaligned latents. We feed the memory and the latents to the transformer (treating them as the 'inputs' and 'outputs' respectively in the notation of [31]). Transformers typically output a value, instead we take the final attention output to be $A_t$. We use three transformer layers, one head and embedding sizes of 128.

MONet outputs $K$ object representations, $z_t$, even when there may be $< K$ objects in a scene. This means that some slots in the object representation do not correspond to an object, we refer to these slots as empty slots. Empty slots may be encoded differently which means that they could be treated as different objects, leading to a memory filled with empty slots. To avoid this we compute which slots in the latent input, $z_t$, are empty using MONet predicted masks, placing a threshold on the sum of pixels, we set this threshold to $4.0$ and found this to work well. In soft alignment, we then mask out those slots to avoid assigning empty objects to slots that are already occupied and to ensure that all empty slots are treated in the same way (rather than creating many new slots for empty slots that may be encoded in different ways). In the hard alignment empty slots are assigned to an $M + 1^{th}$ slot and are not used to update the memory. Additionally, in the hard alignment we ensure that all non-empty slots are assigned to a unique memory slot so that no objects are "lost" in the process of alignment.

## A.3 Transformer + SlotLSTM Transition Module Implementation Details

In Section 2.3 we introduce our slot-wise transition model, Transformer + SlotLSTM. We combine a transformer [31] with a slot-wise LSTM. Our transition model must play two roles (1) At each time-step the model must consider how each object may interact with each other object, (2) the model must be able to integrate information about a single object over time.

For (1) we use transformers. While transformers are often used to process sequences of text, here we use transformers to encode the interactions between an object and all others at that time-step. The input to the transformer are the aligned object latents, $z_t^a \in \Re^{M \times F}$ and the action, $a_t$, taken at that time-step. The action is append to each of the object latents and this is fed to the transformer along with the states of the SlotLSTM (which we detail below).

For (2) we use SlotLSTM this is an LSTM with shared weights applied to each of the $M$ slots in $z_t^a$. Since $z_t^a$ are aligned over time, each LSTM is fed information about a single object over time and

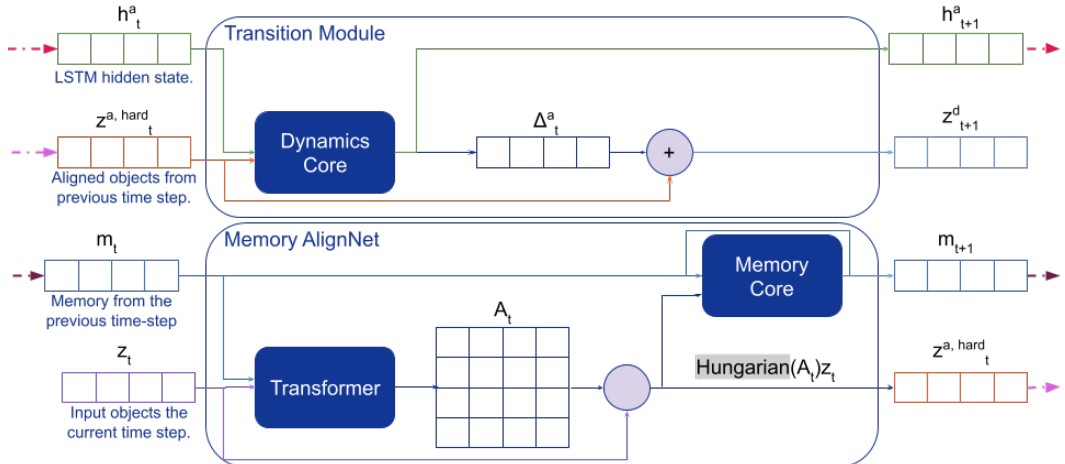

Figure 8: **Details of the Memory AlignNet and its interactions with the Transition module.**

each has its own hidden state (which is passed to the transformer as described above) allowing the model to integrate information about a single object over time. We found that our transition model performed better than regular transformers (Figure 10) we attribute this to the fact that our model has an explicit state for each object which we were able to update over time.

### A.4 Additional Details and Results for the Ablation.

Recall that in the ablation study **only**, we train MONet using ground-truth masks. This means that rather than using an attention network (U-net) to predict masks, $\mu_{k,t}$, we use ground-truth masks and feed these to the slot-wise VAE.

Figure 9 compares models trained using aligned vs. unaligned inputs and pixel-level vs. object-level loss, showing 5 runs per model configuration. Models trained using latent loss can do $10\times$ the number of updates per second compared to models trained using the pixel loss. Additionally, models trained with pixel-level loss tend to become unstable during training leading to high variance between seeds.

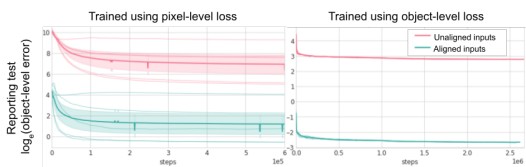

Figure 9: **Comparing models trained using aligned vs. unaligned inputs and pixel-level vs object-level loss.** We see that, across 5 runs, training transition models using aligned inputs and latent-level loss obtains best results.

Figure 10 shows the effect of using different slot-wise transition model cores. We see that a transformer followed by a slot-wise LSTM achieves the best results when training using an object-level loss and aligned inputs.

## B Additional Results on the Playroom Dataset.

In the main body of the paper we demonstrated the alignment modules ability to align object representations over three time-steps. In Figure 11, we demonstrate outputs from the alignment module over 10 steps. The alignment module is able to keep track of objects across time.

Figure 12 shows additional unrolls using OAT, trained under the same conditions as those described in Section A.1. The figure shows consistently good results across many samples.

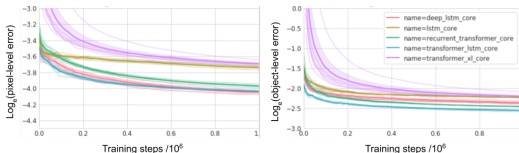

Figure 10: **Comparing transition module cores.** We show five runs for each model configuration. We find that a transformer with slot LSTM outperforms the Transformer XL [5], as well as LSTMs [9], Deep LSTMs and a recurrent transformer that predicts both a state and outputs at each time-step and appends that state to the input to make predictions at the next time-step.

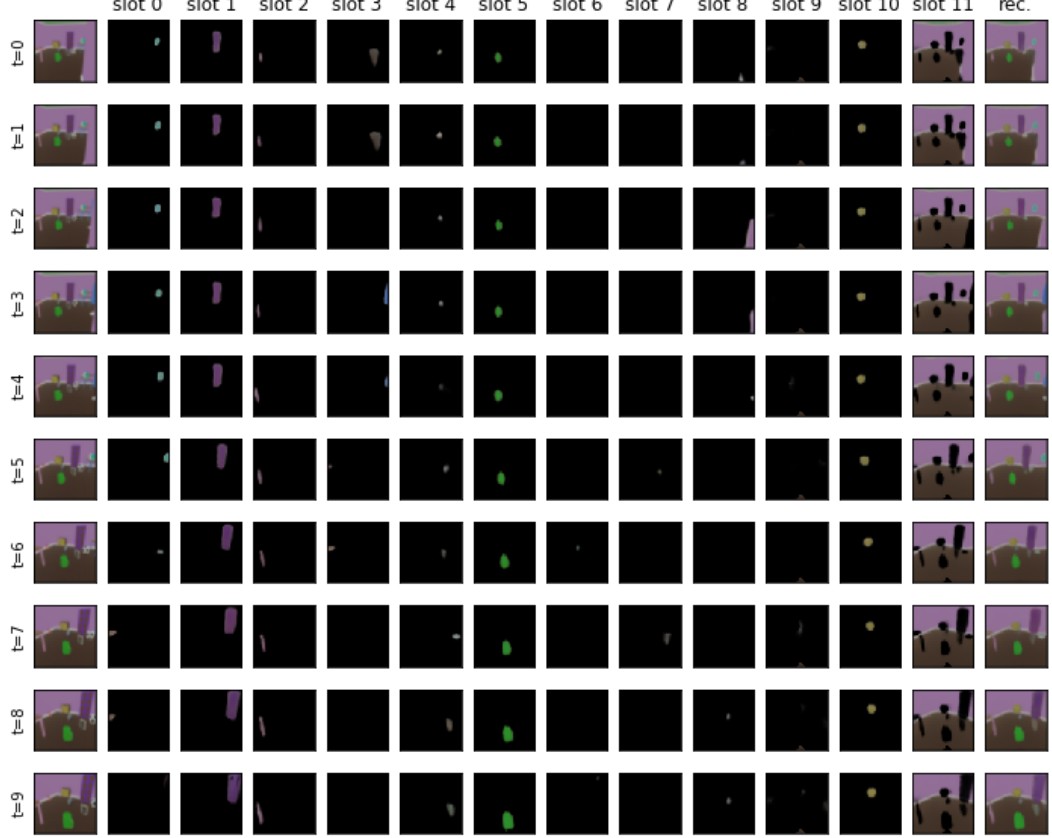

Figure 11: **Aligned inputs (first 4 time-steps) and targets (last 6 time-steps).** Our alignment module outputs object representation vectors which we visualise here using MONet's entity decoder. Most of the objects are now in consistent slots across time, making it easier to compute semantically meaningful losses directly between object representations using a simple L2 loss. Refer to Figure 2 to see how the losses are computed. Notice that while MONet outputs 10 slots, the AlignNet has 12 slots.

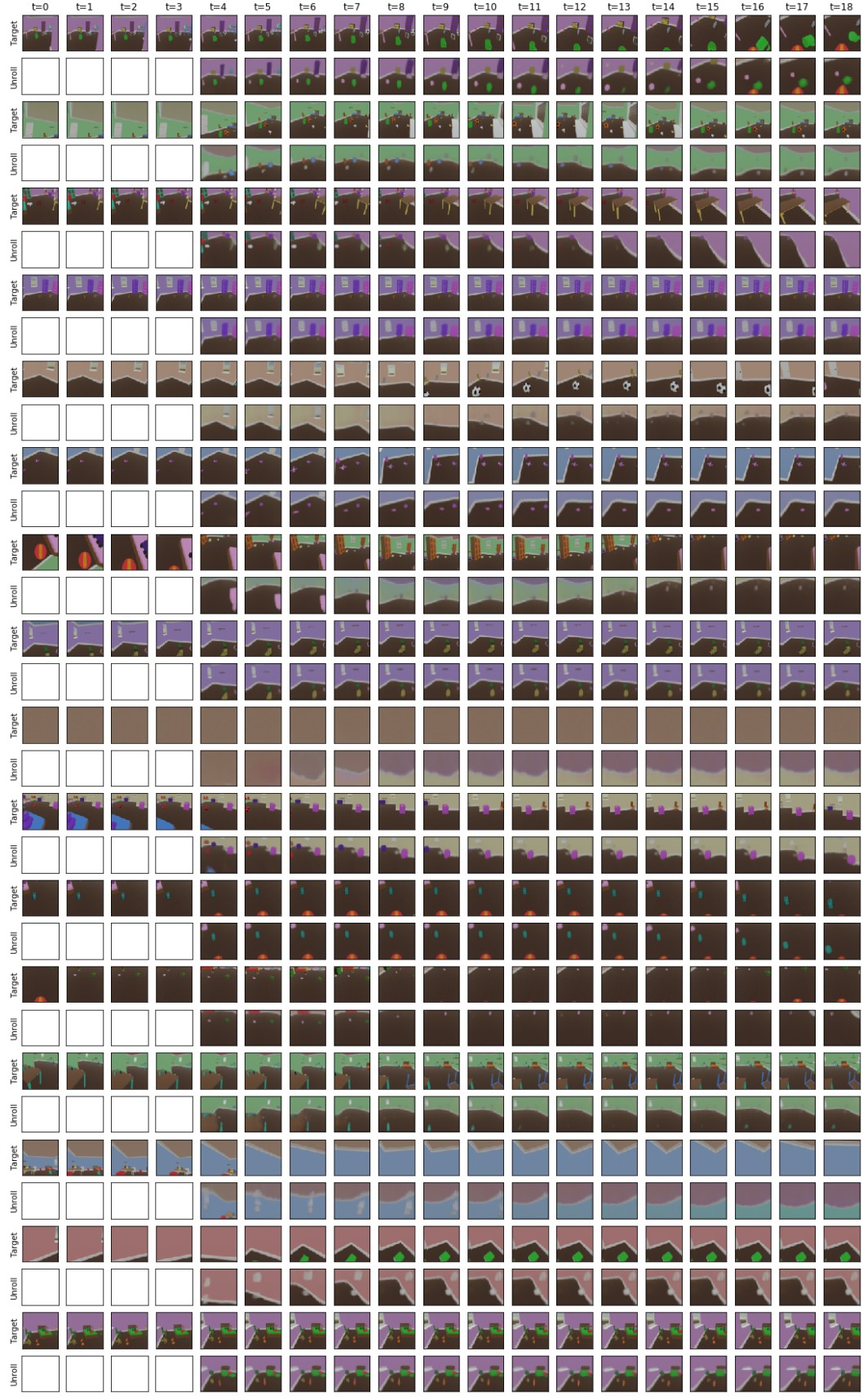

Figure 12: **Additional OAT unrolls.** OAT was trained to take four input steps and unroll for six time-steps, here we demonstrate OAT unrolling for 15 time-steps. Note, this model was trained using the exact same training set-up (detailed in Section A.1 in the Appendix) as results shown in Figure 5 but is a different seed. We see that we get similarly good results across multiple seeds.

In the main text we visualise OAT's unrolls by passing each predicted object representation to MONet's decoder function to reconstruct objects, $\tilde{x}_{t,k}$ and masks $\tilde{\mu}_{t,k}$, and combining these into a single image for each time-step, $\sum_{k=1}^{K} \tilde{\mu}_{t,k}\tilde{x}_{t,k}$. In Figure 13 rather than combining the component objects, we show each of the reconstructed, masked objects, $\tilde{x}_{t,k}\tilde{\mu}_{t,k}$ across time.

## C  Additional Results on the Robotics Dataset.

Figure 14 visualises the MONet outputs on the Robotics dataset. We see that the arm, gripper and objects are each represented in their own object slot, C1 to C7.

Figure 15 shows additional unrolls on the Robotics dataset.

## D  Reproducing OP3 Results

OP3 [32] is a slot-wise scene segmentation and dynamics model. It applies a refinement network (based on IODINE, [10]) to an initial estimate of the scene's slot-wise object features, followed by an action-conditioned prediction model to predict the features at the next time step. The refinement and prediction steps are interlaced through a sequence of steps, making OP3 the closest baseline to OAT.

We tested the correctness of our implementation on the `pickplace_multienv_10k` dataset used in the original OP3 paper and made available by the authors of OP3. The model was trained and evaluated using one refinement step and a next-step prediction in a loop. The results in Figure 16 show that our implementation has learnt the dynamics well enough to cope with the dataset's jumpy object transitions and we are able to reproduce the results similar to those shown in Figure 6(b) of the OP3 paper [32]. Note that we show the result of multiple discrete actions taken over many steps, while they shown only a single discrete action applied at one time step[5]

Following this validation of our implementation, we trained OP3 on the Playroom dataset in a regime that mirrors ours. Four burn-in steps with refinement and next-step prediction allow the model to build its initial estimate of the scene. These are followed by six unroll steps during training, where the refinement is disabled. Here the slot parameters are only updated via the prediction core. For evaluation, we roll-out for 15 steps instead of six to test the dynamics model at long-range prediction. Figure 17 shows eight such roll-out sequences from the model with the best ARI score over the rollout steps.

### D.1  OP3 Hyperparameters

For the results shown in the main section of this paper, we substituted OP3's original relation net-based transition model with a transformer module plus SlotLSTM (transformer + SlotLSTM). This is identical to OAT's setup and allows the fairest comparison. The SlotLSTM's hidden size is also 128. Indeed, results shown in Figure 18 demonstrate that we obtain better performance training OP3 with the transformer + SlotLSTM (introduced in Section 2.3 of our paper), than using OP3's original relation net-based transition model. The transformer uses 2 layers, 4 heads, and embedding size 128.

We train OP3 with the refinement and prediction schedule described above, again similar to OAT. The refinement encoder is a convolutional network with five layers with [64, 128, 128, 256, 256] output channels, kernel shape 5, and stride 1, followed by an MLP with [256, 256] hidden units. The slots have 64 latents each (and hence the MLP outputs 128 posterior parameters). We use stochastic latents only, avoiding OP3's deterministic latents without loss of generality.

The encoder, as in IODINE, is applied slot-wise across the following refinement inputs: the input image, the log likelihood of the image with respect to the predicted output distribution, the current estimate of the slot parameters, the gradient of the log likelihood with respect to the slot parameters, the logits of the decoded object masks, the masks themselves, the gradient of the log likelihood with respect to the masks, and a counter-factual (as in [10]).

---

[5]In Figure 7(a) of the OP3 paper [32] the authors do demonstrate predicting one step into the future, without refinement.

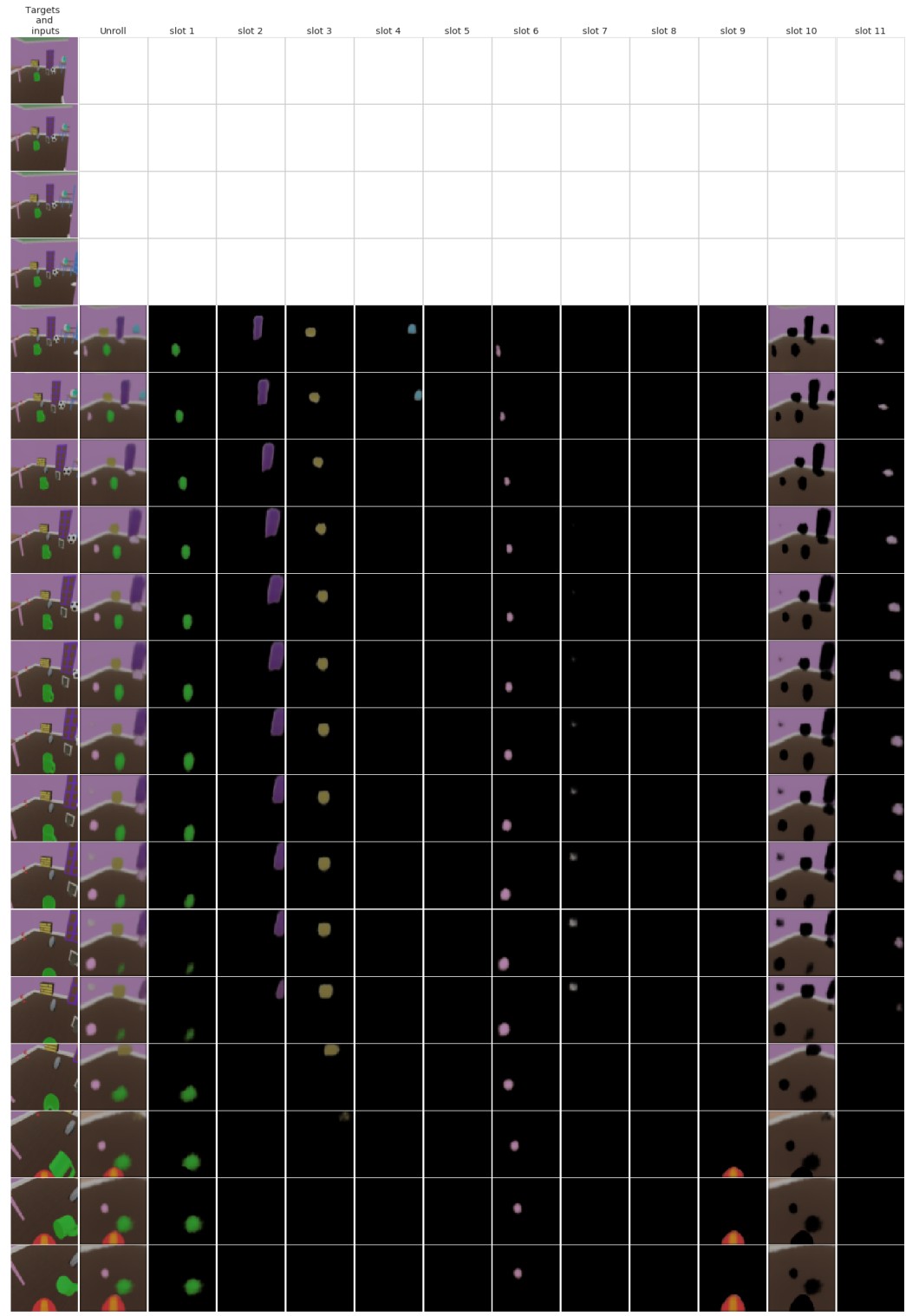

Figure 13: **Showing the components predicted during the unrolls.** Rows correspond to time-steps. Slot one to 11 show the objects predicted during the unroll. OAT sees the first four time-steps and unrolls for the next 15 time-steps.

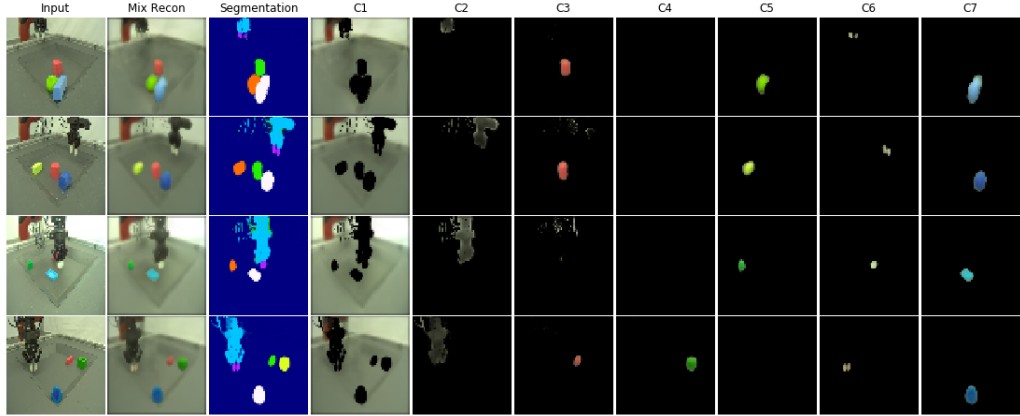

Figure 14: **Scene decomposition and representation module outputs for the real world robotics data.** Each object is correctly placed in its own column.

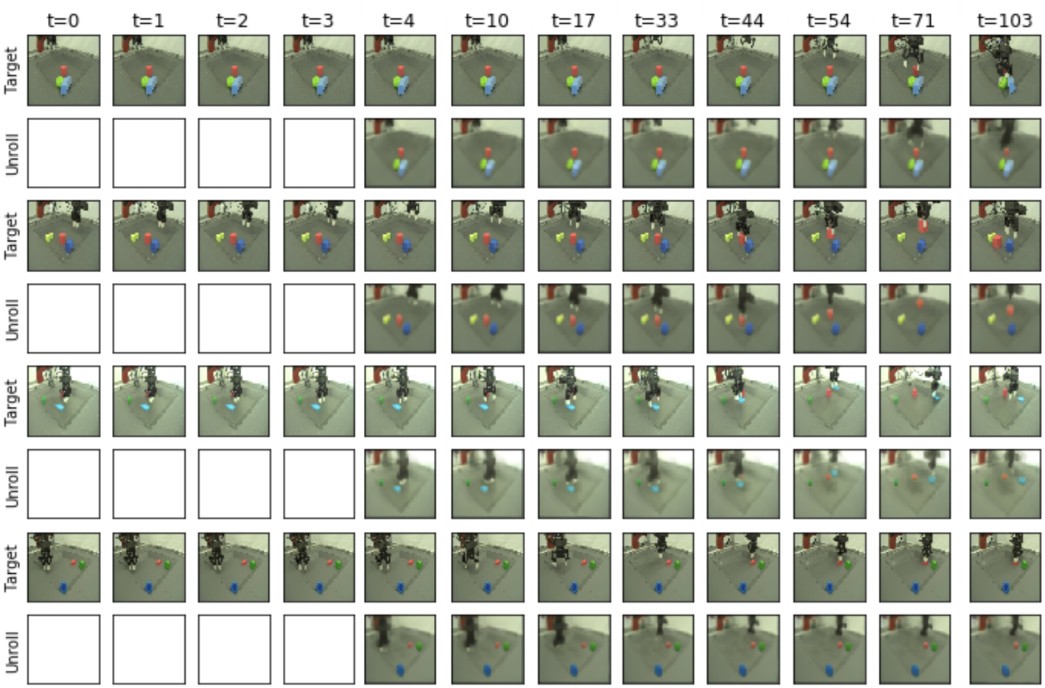

Figure 15: **Unrolling for many more steps than seen during training.** OAT is trained to take four input steps and to unroll for six time-steps, here we demonstrate the model unrolling for 100 time-steps. Impressively, we see that our model learns both to predict the behaviour of the robotics arm accurately and how the arm interacts with the objects. We also see that the model is able to unroll for significantly more steps that seen during training. Note that our model makes prediction in latent space; here we are visualising those latents using MONet's decoder.

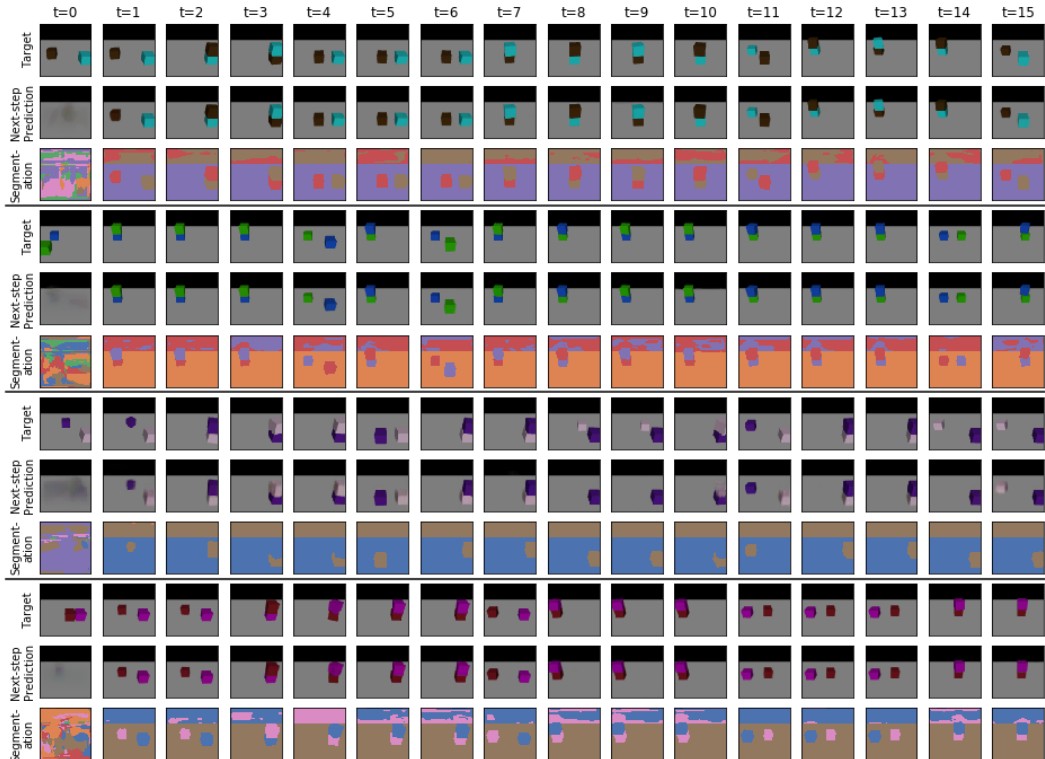

Figure 16: OP3 [32] results on the `pickplace_multienv_10k` dataset released by the OP3 authors. This model was trained as per OP3's standard regime with one dynamics step and one refinement step at each time-step (i.e. no unroll). We've plotted the decoded outputs post-dynamics but pre-refinement at each time-step to show the strength of the transition model. Though the segmentations are imperfect (partly because we used 7 component slots), and the model occasionally drops an object (e.g. sequence 3), it has learnt the dataset's jumpy object transitions perfectly.

The decoder is a broadcast decoder comprising transpose convolutions with [64, 64, 64, 64, 64, 4] output channels, kernel size 5, and stride 1. The decoded mask logits are activated with a tanh scaled by 10.0.

We used a KL loss scale of 0.5 and a fixed output distribution scale of 0.1. To stabilise training, we also clipped gradients to a norm of 5.0. For the refinement inputs, we clipped gradients to a slightly higher norm of 10.0. Finally, we used an effective batch size of 32, the RMSProp optimizer, and a learning rate of 1e-5. We trained all OP3 models for 4 million steps (15 days).

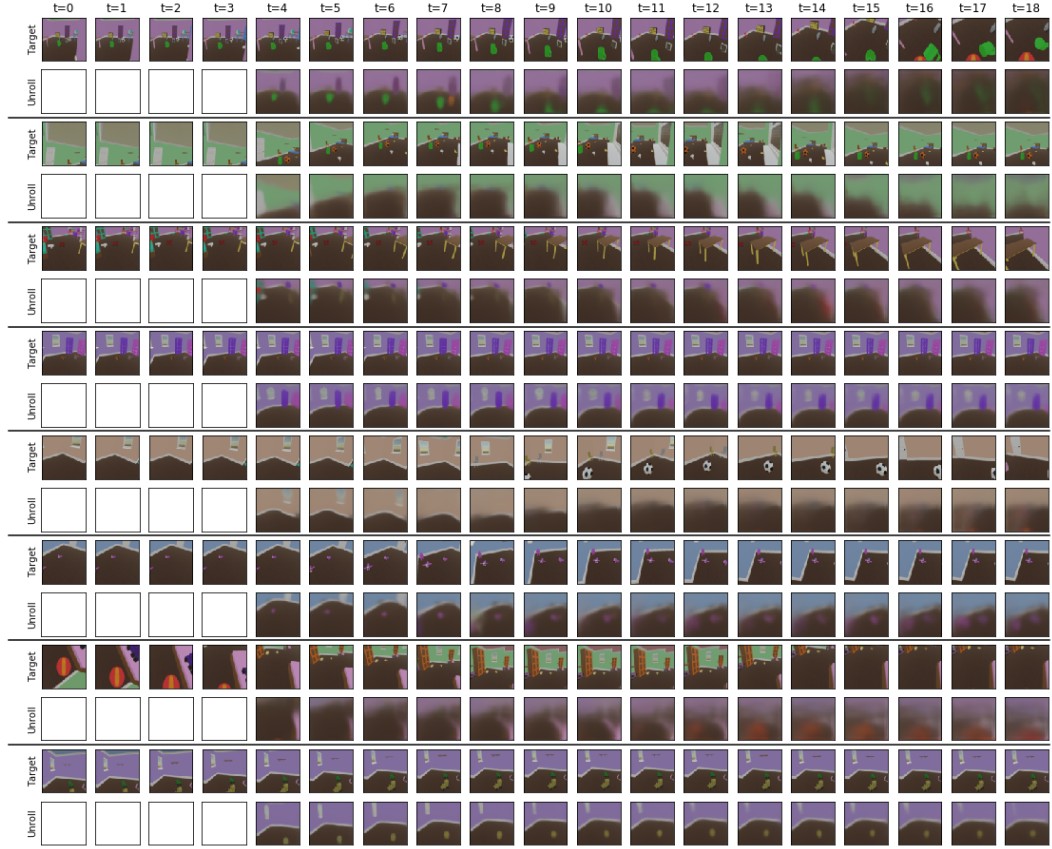

Figure 17: **Baseline OP3 [32] rollouts.** This model was trained with the same number of input steps (four) and unroll steps (six) as our model, OAT (see Figure 5). Here we picked the model with the best ARI score from 10 independent runs. The results are noteworth at t=4 (following the burn-in refinement steps) for accurately placing objects. But for t>4, the prediction core quickly begins to accumulate errors, distorting the size and position of objects and in some cases the floor edges. It further fails to predict the appearance of the avatar (for instance, toward the end of the first sequence), which is an easy-to-learn and predictable consequence of the action space.

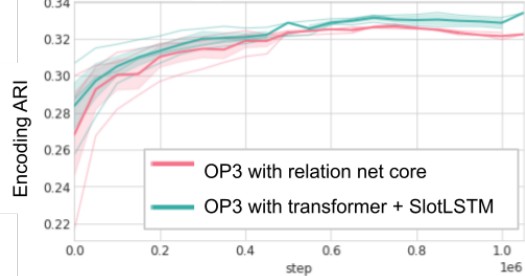

Figure 18: **Comparing OP3 trained with a relation net core vs. a transformer + SlotLSTM core.**