# OpenReview forum: "Unsupervised Object-Based Transition Models For 3D Partially Observable Environments"
_NeurIPS.cc/2021/Conference — NeurIPS 2021 Poster_

### Official Review · Reviewer_Jy59 · 2021-07-16

**Rating:** 7
**Confidence:** 3

**Summary:**

This paper proposes a novel object-based transition model for the decomposition of a scene into objects, followed by alignment to maintain time consistency. It then predicts how these objects will evolve over successive frames. It is trained end-to-end without supervision using transition losses at the level of object-structure representation rather than pixels. A novel alignment module enables the proper handling of two issues that other methods do not do satisfactorily: object persistence and object identity.

**Limitations And Societal Impact:**

Some directions for future work are provided, such as making stochastic predictions about the future and better modeling of uncertainties about objects in the environment.

**Main Review:**

The proposed OAT model (Objects-Align-Transition) has three modules, two of which are the contributions:

- MONet, a scene decomposition and representation module to transform a raw image into a slot-wise object-based representation
- A novel alignment module to ensure that each object is represented in the same slot across space, even if it has disappeared from view
- A novel slot-wise transition model that operates on the object representations to predict future states.

The application of each module is straightforward: MONet creates an object-based representation, the slot-wise transition model uses temporal information to predict future states, and the alignment module ensures that the transition model receives information pertaining to the same object across time. Experiments show that the proposed method outperforms the current state of the art transition-based model, OP3, and that the proposed modules individually contribute to performance improvements. Particularly, the authors pinpoint drawbacks of such model, and how the proposed contributions address these limitations. I would like to have seen comparisons with other methods, especially because there have been a few proposed methods (as acknowledged by the authors) that try to address some of the mentioned issues, and it would have been useful to see to what degree they succeed relative to the proposed method. Figures and other qualitative results are also very helpful in understanding the impact of the proper modeling of object permanence and identity.

The motivation behind the contributions is well explained and related to key issues of current methods. I believe finding ways to address object persistence and object identity would have impact in several field of research, and it is interesting to see progress on that front. The related work section of this paper is well-written, and does a good job positioning the proposed method relative to other published works, showing where they overlap and where the novelty resides, particularly related to the issues of partially aligned environments and semantically meaningful losses. I am not an expert in the field, but the theory and formulations for the various losses and transition models seem solid, with no notable mistakes. Overall, the paper is well-written, clear and concise. As for improvements from previous submissions, I think the paper is self-contained and have no issues with building on top of non-peer-reviewed work for the current submission.

**Time Spent Reviewing:**

2

---

### Official Review · Reviewer_V6Rr · 2021-07-16

**Rating:** 6
**Confidence:** 4

**Summary:**

This paper focuses on decomposing the scene into objects and predicting how these objects evolve in successive frames. This method is based on MONet and improves object alignment by proposing AlignNet. Compared with OP3, the quantitative results show better performance.

**Limitations And Societal Impact:**

It seems that many of the claims made in the paper about the method's ability to process part of the observable scene and capture the reappearance of objects are not fully supported by the experimental results.
- The authors claim that the proposed method is better in dealing with 3D partially observable environments. In view of the fact that the image is too small to study, I think it is better to perform a quantitative detailed analysis: given the different ratios of observations (such as 10%, 50%, etc.), show the robustness of the method.
- The authors mentioned that methods like C-SWM "is unlikely to scale well to partially observable environments with significant movement of objects across the field of view". But there are no quantitative or qualitative results to prove this statement.

**Main Review:**

I thank the authors for their efforts to rewrite the paper and include the AlignNet method. I saw a few strengths in this paper: Compared with pixel-level loss, the author demonstrated better performance using object-level loss. In addition to proposing the transformation module, the authors also compared the core of different transition models and found the best one using transformer followed by LSTM. I am inclined to accept, but it seems that many of the claims made in the paper about the method's ability to process some observable scenes and capture the reproduction of objects are not fully supported by the experimental results (explained in detail in the next section). I would like to hear the authors' response to further verify my assessment.

After reading the rebuttal, most of my concerns were resolved. I thank the authors for their response.

**Time Spent Reviewing:**

4

---

> ### Author Response · Authors · 2021-08-09
> **Response to Reviewer V6Rr**
>
> Thank you for taking the time to read our paper and for your question and suggestion. Please find our response below:
>
> > It seems that many of the claims made in the paper about the method's ability to process part of the observable scene and capture the reappearance of objects are not fully supported by the experimental results.
>
> The main claims of the paper are to deal correctly with object persistence and object identity (which directly aids the model in dealing with reappearance). Dealing correctly with object persistence and object identity allows us to feed objects in consistent slots to the transition model and to compute object-level losses and we demonstrate the importance of each of these in Section 4.2 (Figure 6 and Table 2). We also demonstrate the role of alignment in transition models by showing that our model, OAT, performs better when using alignment than without, OT (see Figure 1 OT vs OAT).
>
> We demonstrate that the model can deal with reappearance both on the robotics dataset in Figure 7 and in Section 4.2, Figure 6 (the ablation) we showed that only when training with aligned latents is it possible for the model to learn reappearance.
>
> Additionally, when re-writing the paper (since our previous submission) we removed the emphasis on reappearance, only mentioning it once in the methods section and at multiple points where relevant in the results section. The main contribution of our paper is to deal with object persistence and object identity and the improvements this leads to when training object-based transition models.
>
> > The authors mentioned that methods like C-SWM "is unlikely to scale well to partially observable environments with significant movement of objects across the field of view". But there are no quantitative or qualitative results to prove this statement.
>
> Thank you for this comment. This is indeed why we use the term “unlikely”. However, we would be happy to revise this statement.

---

### Official Review · Reviewer_gBUs · 2021-07-17

**Rating:** 6
**Confidence:** 5

**Summary:**

Summary: The paper proposes a slot-based memory architecture to factorize the visual world into a set of objects to preserve object persistence and object identity. Object persistence is important to make sure that the representation of objects are in the memory even though the objects are no longer perceptible, and object identity is important to make sure the same slot refers to the same object at later time-steps.

The paper conducts experiments on two sequential image datasets, and achieve better results as compared to the state of the art object centric transition models.

**Ethical Concerns:**

No ethical concerns.

**Limitations And Societal Impact:**

Please refer to main review for pointers for improving the presentation of the current work.



**Main Review:**

Architecture: The proposed architecture consists of 3 components
(a) Representation module: a module for parsing the visual world into set of slots. For this component paper proposes to use MONet.
(b) Alignment module: the goal of this module is to align objects across slot representations at different time-steps.
(c) Slot-wise Transition module: The paper proposes to use SlotLSTM where weights are shared across object slots and can be instantiated in many different ways.

Pros:
- I appreciate and like the idea of evaluating slot-wise transition models for partially observable 3D environments. Different papers have argued before ([1, 2]) of the importance of using structured representations in 3D environments, but I'm not aware of any work which evaluates such ideas in a systematic way on such environments.
- The paper is easy to read, and generally well written.
- I acknowledge the importance of doing analysis in section 4.2. Its a useful result that the object level loss performs better as compared to pixel-level prediction (for concreteness object-level loss i.e., a loss which is computed directly between predicted and target object representations (as compared to a pixel-level loss).

Cons:
- I'm concerned about the depth of the experiments, as well as the statements regarding the relevant literature. The authors missed the relevant literature which tries to achieve *both* object persistence and identity (though they don't evaluate on 3D partially observable environments)

*Comments about the contributions*:

- When I read the title of the paper, I was pretty excited because the paper seems to be one of the very few papers which tries to use factorized representations for partially observable 3D environments. Though, on reading the introduction it seems that the paper proposes a new architecture for learning slot-wise transition model, and just evaluates the proposed architecture on 3D environments, so IMO title seems to be not aligned as to what paper is trying to do (in its current form as of now)

*two issues that are not handled satisfactorily by other transition models, namely object persistence and object identity*

I'm slightly concerned about this statement. This statement is factually incorrect. There's some recent work which tries to achieve both object persistence and object identity in a single architecture. For ex. See SCOFF [1] and its followup Neural production systems [2]. SCOFF consists a set of slots (or object files) where each slot attends to different part of the state space (i.e.. different entity). Each object file is parameterized as a LSTM. Different slots are represented by different LSTM (with dynamic weight sharing). In SCOFF,  the state of object file persists over time (to address object identity), and different object files attend to different parts of the state space via top-down attention where the "query" is a function of the hidden state of the object file.  So, in essence the Slot-wise Transition module (set of LSTMs with weight sharing) used in the proposed work is a baseline in SCOFF.  In NPS, authors also claim to achieve better results as compared to OP3. That said, both in SCOFF or NPS, the authors rather evaluate such ideas on toy 2D domains or for "easy" downstream RL task.

References:
- [1] The ThreeDWorld Transport Challenge: A Visually Guided Task-and-Motion Planning Benchmark for Physically Realistic Embodied AI
https://arxiv.org/abs/2103.14025
- [2] CausalWorld, https://arxiv.org/abs/2010.04296
- [3] Object Files and Schemata: Factorizing Declarative and Procedural Knowledge in Dynamical Systems,
https://arxiv.org/abs/2006.16225 (ICLR'21)
- [4] Neural Production Systems (https://arxiv.org/abs/2103.01937)

*Comments about the experimental results*:

- The authors evaluate the proposed architecture using three downstream metrics: Encoding ARI, Unroll Pixel Error and Unroll ARI. In Table 1, paper shows the importance of proposed alignment module.  It would be useful to know to variance of both the baselines and proposed model across different seeds. I understand that such models can be a bit unstable, so it would be helpful to know the variance and standard deviation across seeds (the authors refer to Appendix for OP3, but it would also be useful for OAT).

- The results mentioned in the section 4.3 are encouraging but it would also be useful to report results on some downstream metric i.e., does learning the representations using the proposed method leads to better downstream performance on some RL task ? Otherwise its hard to judge the significance of these results.

Questions for authors:

I'm pretty happy to change my ratings if authors can address these points in a satisfactory way:

- *Acknowledge relevant literature*: The reviewer appreciates the effort of trying to use factorized representations for 3D partially observable environments but its importance to acknowledge the relevant literature. OAT consists of 3 modules (a) MONet (b) AlignNET (c) SlotLSTM. Both MONet and SlotLSTM (like in SCOFF/NPS) are proposed and used in literature already so it would be useful to acknowledge that there exists work which has the property of both object persistence and identity and mention the limitations of previous work i.e., it does not evaluate on 3D partially observable environments.

-  *Performance of the proposed method on downstream RL tasks*:vIt would be useful to report downstream metric for the robotics experiment. It would be very interesting to see, if the representations learned by the proposed model leads to more powerful policies or improved sample efficiency on the downstream RL task.

-  *Performance of the proposed method on downstream RL tasks for Playroom env*:  The paper mentions that the data  is collected from a
pre-trained agent moving in a simulated 3D environment. It would again be useful to know if the learned representations lead to better generalization (or improved sample efficiency or transfer) for downstream RL task.

-  *Performance of the proposed method on 2D task*: It would be useful to know how the representations learned by the proposed method are useful as compared to [1]. In order to evaluate this, authors can follow the similar experimental procedure as in [1] i.e. train OAT on sequence of frames, and use the learned representation of entities as input to Transformer (as compared to MONet used in [1]). Does the representations learned by the proposed method leads to improved downstream results ?

[1] Object-based attention for spatio-temporal reasoning: Outperforming neuro-symbolic models with flexible distributed architectures, https://arxiv.org/abs/2012.08508

Please don't hesitate to ask for any clarifications, if something is not clear.

**Time Spent Reviewing:**

5 hours

---

> ### Author Response · Authors · 2021-08-06
> **Response to Reviewer gBUs.**
>
> Thank you for taking the time to read our paper and for your suggestions. Please see our response below.
>
> > *two issues that are not handled satisfactorily by other transition models, namely object persistence and object identity*. I'm slightly concerned about this statement. This statement is factually incorrect. There's some recent work which tries to achieve both object persistence and object identity in a single architecture. For example. See SCOFF [1] and its followup Neural production systems [2].
>
> Although this hinges on the interpretation of the word “satisfactorily”, we will adjust this sentence to acknowledge the two papers the reviewer highlights. (We also note that NPS is concurrent with our work.)
>
> Please also note that SCOFF and NPS do not present an action-conditional transition model. Rather they present a component that could potentially be used in a transition model.
>
> > Acknowledge relevant literature: The reviewer appreciates the effort of trying to use factorized representations for 3D partially observable environments but its importance to acknowledge the relevant literature. OAT consists of 3 modules (a) MONet (b) AlignNET (c) SlotLSTM. Both MONet and SlotLSTM (like in SCOFF/NPS) are proposed and used in literature already so it would be useful to acknowledge that there exists work which has the property of both object persistence and identity and mention the limitations of previous work i.e., it does not evaluate on 3D partially observable environments.
>
> Thank you for suggesting these additional papers, they are indeed very interesting and consider similar problems to our own paper. As promised above, we will include both SCOFF and NPS in the revised version of our paper. On an additional note we also propose a Transformer + SlotLSTM architecture for the transition model (Section 2.3, Line 134) and show this to be better than just a SlotLSTM (Figures 10 in the Appendix). Below we offer a comparison between our model and SCOFF/NPS, which we can include in our paper.
>
> - Both SCOFF and NPS place a heavy emphasis on interpretability of the “rules” which are applied to objects, which is not the focus of our work.
> - While SCOFF shows certain schemata being applied to certain object files, there is no clear decodable representation for the objects. In our model we have three distinct steps that represent the objects, align them and then apply the transition model. This means that we can (1) visually verify slot (identity) consistency (see Figure 11) and (2) visualise each individual object at each time-step during the unroll (see Figure 13 in the Appendix).
> - SCOFF also assumes that there are multiple objects of the same “class” or “type” that share dynamics. In our paper we relax this assumption and have a single transition model acting on all objects.
> - In our work we demonstrate results in more complex 3D environments; we show that OAT produces accurate unrolls for 100 times on a Real World Robotics dataset, while SCOFF shows results on a 2D dataset.
> - NPS uses “rules” to decide on how slots should interact with one another, while we take a more general approach and use a transformer to determine slot interactions.
>
> > Performance of the proposed method on downstream RL tasks: It would be useful to report downstream metric for the robotics experiment. It would be very interesting to see, if the representations learned by the proposed model leads to more powerful policies or improved sample efficiency on the downstream RL task.
>
> Thank you for this suggestion. We would like to draw attention to the fact that the focus of our model is not only on representation learning, but rather on learning good object-based transition models which we imagine could be used in future work for planning. Incorporating object-based planning into agents is an open research area and would be a significant research project, hence this is outside the scope of the present paper. However we will consider this in future work.
>
> > Performance of the proposed method on downstream RL tasks for Playroom env: The paper mentions that the data is collected from a pre-trained agent moving in a simulated 3D environment. It would again be useful to know if the learned representations lead to better generalization (or improved sample efficiency or transfer) for downstream RL task.
>
> Again, this is beyond the scope of the present paper.
>
> > Performance of the proposed method on 2D task: It would be useful to know how the representations learned by the proposed method are useful as compared to [1]. In order to evaluate this, authors can follow the similar experimental procedure as in [1] i.e. train OAT on sequence of frames, and use the learned representation of entities as input to Transformer (as compared to MONet used in [1]). Does the representations learned by the proposed method leads to improved downstream results?
> [1] Object-based attention for spatio-temporal reasoning: Outperforming neuro-symbolic models with flexible distributed architectures, https://arxiv.org/abs/2012.08508
>
> Thanks. This is another great suggestion for future work.

---

> > ### Comment · Reviewer_gBUs · 2021-08-11
> > **Thanks for responding :)**
> >
> > The reviewer appreciate the author's comments.
> >
> > **but rather on learning good object-based transition models which we imagine could be used in future work for planning**
> >
> > If the focus on learning object-based transitions model, which can be used for downstream task, as of now no downstream task is considered in this paper, so its hard for the reviewer to evaluate whether the representations learned are useful or not.
> >
> > **Thanks. This is another great suggestion for future work**
> >
> > I'm not sure why this experiment is difficult to do (2D task). I'm *happy* to increase my score as long as authors provide the performance on at-least one downstream task.
> >
> > 2D task seems the easiest to me (given the fact that the proposed method already relies on MoNet), so if the representations are indeed learning something meaningful it should be reflected in downstream performance. In order to benefit your method, you can even construct a simple task such that their's some occlusion or some violation in intuitive physics (for ex. like in [1], [2]).
> >
> > - [1] IntPhys: a benchmark for visual intuitive physics reasoning, https://arxiv.org/abs/1803.07616
> > - [2] Modeling Expectation Violation in Intuitive Physics with Coarse Probabilistic Object Representations https://proceedings.neurips.cc/paper/2019/file/e88f243bf341ded9b4ced444795c3f17-Paper.pdf
> >
> > ===================
> >
> > These points are something which can be easily fixed, so I trust authors, that they will do a good job of re-emphasizing what their actual contribution is. My rating is not effected by how relevant work is mentioned (and only by the absence of performance on any downstream task).
> >
> > **Please also note that SCOFF and NPS do not present an action-conditional transition model. Rather they present a component that could potentially be used in a transition model**
> >
> > Good point. Though, as the authors mentioned, their focus is on the representation and the introduction very heavily relies on saying that the proposed method is the only way which allows aligning the representations of different objects. So, if this is the focus of the work, then probably this is incorrect in the light of SCOFF.
> >
> > **SCOFF also assumes that there are multiple objects of the same “class” or “type” that share dynamics. In our paper we relax this assumption and have a single transition model acting on all objects.**
> >
> > This would be the baseline for SCOFF i.e., all the slots share the same schemata (i.e., GRU update function), so I don't see how the paper "relax" this assumption. Imagine you have 3 objects, (a) blue ball (b) red ball (c) green square. You would probably assume that blue and red balls follow the similar dynamics and green square would follow different dynamics.
> >
> > **While SCOFF shows certain schemata being applied to certain object files, there is no clear decodable representation for the objects. In our model we have three distinct steps that represent the objects, align them and then apply the transition model**
> >
> > I'm not sure I agree. I could be misunderstanding the SCOFF work. But different slots do cross-attention on the image, and then they have a decoder (which one can use to decode slot-wise like in MoNet). I agree with the authors though, this is not visually shown in SCOFF.

---

> > > ### Author Response · Authors · 2021-08-26
> > > **Results on a downstream robotics task.**
> > >
> > > We are very happy to report that, following the reviewer’s suggestion, we have now succeeded in demonstrating the effectiveness of OAT on a downstream task. Specifically, we have applied OAT in a simulated robotics setting involving a robot arm that is trained to stack objects. We have compared performance using OAT to a MONet baseline, and we were happy to find that the OAT version indeed outperformed the baseline.
> > >
> > > In a little more detail, we compared agents trained using the following representations:
> > >
> > > * OAT: We use aligned object representations from a pre-trained OAT model.
> > > * MONet: We use MONet representations from a pre-trained MONet model.
> > >
> > > To obtain these representations we use OAT and MONet models trained on synthetic data using the exact same hyper-parameters. Both OAT and MONet produce good segmentations, and the unrolls from OAT are similar to those in the paper on real robotics data. We then freeze the parameters of OAT and MONet before training the agent. Our agents are trained to stack one object on top of another and are given intermediate rewards for reaching, grasping, lifting and placing objects. This means that the agent is rewarded for solving easier tasks alongside the more complex ones.
> > >
> > > We trained two agents using the two representations listed above using three different seeds (6 agents trained in total). We found that as the sub-tasks got more difficult - reaching, then grasping, then lifting, then placing - the representations from the OAT model led to increasingly large gains in performance over representations from MONet. We now give more details, with the tasks ranked by difficulty from reaching to stacking.
> > >
> > > *Easy tasks*
> > >
> > > **Reaching**: The models have converged and we see that agents trained with both representations, across all three seeds, perform at a similar level. Please see the reaching rewards for each seed below:
> > >
> > > |  |  Reaching|reward|
> > > | ---- | ------ | -------- |
> > > | seed  | OAT | MONet |
> > > |7 |  258  | 188 |
> > > |13   |  301 | 100  |
> > > |21   |  175  | 167 |
> > >
> > > **Grasping**: We see that agents trained using OAT representations take off much earlier in training than agents trained using MONet representations and we see agents trained using MONet representations plateau earlier than those using OAT representations. Please see the grasping rewards for each seed below:
> > >
> > > |             |  Grasping | reward |
> > > | ---------- | --------- | ------------- |
> > > | seed     | OAT     | MONet    |
> > > |7            |  129     | 65           |
> > > |13          |  267     | 56           |
> > > |21          |  64       | 62           |
> > >
> > >
> > > *More challenging tasks*
> > >
> > > **Lifting**: After training for 20k episodes, we see that for 2/3 seeds, agents trained using OAT representations have taken off, while agents trained using MONet representations have not. Please see the lifting rewards for each see below:
> > >
> > > |              |  Lifting | reward      |
> > > | ----------- | -------- | --------------|
> > > | seed     | OAT     | MONet    |
> > > |7            |  100     | 7             |
> > > |13          |  120     | 3             |
> > > |21          |  16       | 2             |
> > >
> > >
> > > **Place wide and narrow**: Similar to lifting results, we see that for 2/3 seeds agents trained using OAT representations have started to perform the task, while agents trained using MONet representations have not. Please see the place wide and narrow rewards for each seed below:
> > >
> > > |    Place   |  narrow | reward   |
> > > |----------- | -------- | -------------------- |
> > > | seed     | OAT     | MONet           |
> > > |7            |  31      | 1                      |
> > > |13          |  52      | 1                      |
> > > |21          |  3        | 0                      |
> > >
> > > |   Place  |  wide | reward   |
> > > |----------- | ------- | ------------- |
> > > | seed     | OAT     | MONet   |
> > > |7            |  9        | 0             |
> > > |13          |  24      | 0             |
> > > |21          |  0        | 0             |
> > >
> > >
> > > **Stacking**: Only one agent trained using OAT has started to perform this task, reaching a reward of 7. None of the agents trained using MONet representations have been able to perform this task.
> > >
> > > Given the quality of these results, we hope the reviewer will increase their score, as promised.

---

> > > > ### Comment · Reviewer_gBUs · 2021-08-26
> > > > **Thanks for running extra experiments.**
> > > >
> > > > I appreciate that the authors took time and ran extra experiments. I've increased the score to 6. That said, I am assuming that authors  would rephrase their introduction (object persistence and object identity) as mentioned in my review.
> > > >
> > > > One clarification: I see that the three seeds used are 7, 13, 21. Did authors ran experiment with these 3 seeds only ? Or authors ran experiments with many seeds and then selected topk (k=3) seeds ?

---

> > > > > ### Author Response · Authors · 2021-08-26
> > > > > **Thank you.**
> > > > >
> > > > > Thank you for increasing your score. We will indeed rephrase the introduction.
> > > > >
> > > > > > One clarification: I see that the three seeds used are 7, 13, 21. Did authors ran experiment with these 3 seeds only ? Or authors ran experiments with many seeds and then selected topk (k=3) seeds ?
> > > > >
> > > > > We ran experiments with **only** these three seeds.

---

> > > > > > ### Comment · Reviewer_gBUs · 2021-08-26
> > > > > > **Thanks for the clarification.**
> > > > > >
> > > > > > Great! I'm glad to hear that authors ran experiment as suggested by the reviewer.

---

### Official Review · Reviewer_sozU · 2021-07-18

**Rating:** 8
**Confidence:** 4

**Summary:**

The paper presents an approach to learning implicit representations of
(simple synthetic) 3D scenes from a few consecutive video frames.
Based on the Monet scene decomposition and representation approach,
the new proposed approach adds: 1) a differentiable alignment module
(Hungarian algorithm based) that helps maintain the same scene component in the
same model representation slot, and 2) a temporal transition component that
links representations, 3) loss based over object-representations instead of over pixels.
In comparison to a state-of-art approach (OP3), a better representation is achieved and
is better maintained many frames past training.


**Limitations And Societal Impact:**

The paper could be more critical about what was not working well, and how far the research ahs to go to be practical.

**Main Review:**

The approach is built on 3 components: the previous Monet component, a
new alignment component, and a new temporal transition component that links
consecutive time frames. The paper clearly explains the motivation and
theory behind each component, and experiments demonstrate the benefits
from the components.
Generally, the paper is quite clear, although many of the images depend on being able to
zoom the paper, and using higher resolution embedded images would have helped.
The paper does not give a precise definition of the experimental 'unroll' step, so it
is a little hard to evaluate how much information is given as input into the unroll steps.
While the proposed approach is good theoretical and experimental improvement, the
reconstruction images show that there are still many extra and missing scene
components. This could have been discussed. The quantitative results only used the
foreground objects (which seems reasonable given the majority of the pixels are the
background and this seems to be reasonably represented) and the numerical results are
much improved, but the high numbers did not relate to my visual inspection of the
reconstruction results (fig 1 good, fig 5 top reasonable, others not so good).
Perhaps this is a consequence of the ARI measure being based on the inferred object masks,
and none of these seem to have been shown in the paper or supplementary materials.
The pixel error also seems to include the background, which is well represented and so
dominates the error calculation.

In summary: several clearly beneficial theoretical and practical innovations
giving a substantial improvement to the implicit learning from video research theme,
but the research theme is still in an early stage.


**Time Spent Reviewing:**

3

---

> ### Author Response · Authors · 2021-08-09
> **Response to Reviewer sozU**
>
> Thank you for a positive review and for your constructive questions and feedback.
>
> > Generally, the paper is quite clear, although many of the images depend on being able to zoom the paper, and using higher resolution embedded images would have helped.
>
> Thank you for this feedback. To make the figures more clear we can remove (additional) intermediate time-steps and try to reduce the white spaces between around the images so that they can be made larger. We have also included additional (larger) images in the appendix.
>
> > The paper does not give a precise definition of the experimental 'unroll' step, so it is a little hard to evaluate how much information is given as input into the unroll steps.
>
> Thanks for this question. The unroll steps are defined by the equations in Section 2.3 lines 128-129. When evaluating the model, in the unroll phase the model does not have access to the input image; it simply takes the previous outputs from the dynamics model, along with the hidden state and an action and the transition model predicts the next representations. We will update the paper to make this more clear.
>
> > While the proposed approach is good theoretical and experimental improvement, the reconstruction images show that there are still many extra and missing scene components. This could have been discussed.
>
> Thank you for this comment. Please note that in Section 4.1 lines 258-261 we say that: “In some of  the examples the targets appear to have more objects than those seen in the unroll: this is because the model has only seen the first four frames and has not seen those other objects in the room and therefore does not have enough information to predict where unseen objects will appear”.
>
> The "extra" objects are likely to arise from cases where only part of the object was visible in the first four time-steps and so appears to be an "extra" object but is actually just the part of an object that was visible in the first four frames.
>
> > The quantitative results only used the foreground objects (which seems reasonable given the majority of the pixels are the background and this seems to be reasonably represented) and the numerical results are much improved, but the high numbers did not relate to my visual inspection of the reconstruction results (fig 1 good, fig 5 top reasonable, others not so good).
>
> Thank you for this question. Given that the model only sees the first four frames and therefore cannot know what is around the corner, the predictions in Figure 5 are very good. For example in the bottom example in Figure 5 it is hard for the model to tell that the leg belongs to a table because it has not seen the rest of the table, but it is able to correctly predict where the green cupboard is and when it goes out of view, similarly with the red object.
>
> > Perhaps this is a consequence of the ARI measure being based on the inferred object masks, and none of these seem to have been shown in the paper or supplementary materials.
>
> All of the object segmentations that we show are the reconstructed object image multiplied by the reconstructed mask image which gives the images their black background. We are happy to add additional images of the masks to the paper.
>
> > The pixel error also seems to include the background, which is well represented and so dominates the error calculation.
>
> Yes, thank you for this comment. We considered different variations for this loss however pixel loss is the most commonly used loss in the literature. Further, we also report object-error when possible. However it is only possible to accurately compute object error when using GT masks, otherwise we have to compute a lower bound on the error using the Hungarian.

---

> ### Comment · Reviewer_sozU · 2021-08-28
> **summary**
>
> Based on the other reviews and author responses, I propose to keep the same score. I encourage the author(s) to make the proposed improvements in the final version (that others will cite).

---

### Official Review · Reviewer_vmwV · 2021-07-19

**Rating:** 7
**Confidence:** 4

**Summary:**

This paper faces a relevant challenge to develop intelligent agents, this is, accounting for object permanence. The key idea is to align objects across time using a slot-based memory. Specifically, the proposed model, Objects-Align-Transition (OAT), consists of 3 main steps: i) Objects: scene is decomposed into objects using an existing technique (Monet), ii) Align: relevant objects are mapped to a slot-wise memory keeping object identity and dealing with object permanence (identity and persistency), iii) Transition: an object transition model uses a "transition prediction loss" to predict future states, maintaining object identity across time. As a result, the method is able to track objects across time by mapping each object into a slot-wise memory and mapping subsequent views of the same object to the same slot.  As relevant features: i) Model is trained end-to-end without supervision, and ii) Transition loss operates at the level of an object-structured representation as opposed to an unstructured space, such as pixels. Results using synthetic and real constrained scenarios indicate that the model is able to handle object persistence and object identity, dealing with problem such as object occlusion and re-appearance in partially observable environments.

**Ethical Concerns:**

Not ethical issues

**Limitations And Societal Impact:**

Not potential negative societal impacts.

**Main Review:**

Overall the paper is clear and easy to follow. The main contributions are clearly stated and ablation experiments demonstrate their usefulness. Main concern is the potential lack of applicability of the proposed method to more complex scenarios. The current setup only includes simplified scenarios with few objects and limited dynamics. Real natural scenes are harder to decompose into objects (or fully represent as a set of objects). Also, more objects require more memory slots, especially if complex objects have to be further decomposed into several parts to fully model their dynamics. A related point is that the more objects there are, the more difficult it will be to align them. It's entirely possible that a model with a global representation and a "pixel space" loss will perform better in the suggested -hypothetical- case. However, as the experiments show, in the limited environment considered here, the proposed model shows good performances and it is able to outperform alternative approaches. In this sense, this work is part of initial efforts to develop models that can tackle problems such as object permanence. This is crucial to deploy embedded agents in natural environments, as a consequence, I recommend its acceptance to Neurips.

Further comments:
Is the model able to capture some intuitive physics when learning the transition model?
What is the distance metric in the the Adjacency Matrix used to apply the Hungarian algorithm?
Is the method able to solve false object detections by Monet?
Can you clarify how the model is able to improve out-of-distribution generalization or few-shot transfer?
Figures are difficult to see, frames are really small making very hard to appreciate the results.
There are problems with references, several of them do not include information about publication venue.
Missing reference: Tracking the world state with recurrent entity networks.

**Time Spent Reviewing:**

3.0 hours

---

> ### Author Response · Authors · 2021-08-09
> **Response to Reviewer vmwV**
>
> Thank you for a positive review and for your constructive and interesting questions. Please find our response below:
>
> > Main concern is the potential lack of applicability of the proposed method to more complex scenarios. The current setup only includes simplified scenarios with few objects and limited dynamics. Real natural scenes are harder to decompose into objects (or fully represent as a set of objects). Also, more objects require more memory slots, especially if complex objects have to be further decomposed into several parts to fully model their dynamics.
>
> Thank you for this comment. In our paper, a key innovation, as you say, is dealing correctly with object persistence and identity, rather than focusing on scene decomposition. For scene decomposition, we use MONet. However, we imagine that as (unsupervised) scene decomposition models improve, these could be substituted into OAT in place of MONet.
>
> Further, we use an environment that is more complex than is typically used in object-based transition models e.g. in OP3 or C-SWM since it involves an embodied agent moving in a 3D Playroom environment. Additionally, existing object-based transition models struggle to model dynamics even in the “simplified scenarios” and our model is among the first to show strong results in a more complex environment.
>
> > What is the distance metric in the Adjacency Matrix used to apply the Hungarian algorithm?
>
> We use the L2 loss between all pairs of object representations. This is mentioned in Section 3.2 on Line 320. We also use the L2 to compute the errors for Table 2. We will update the text to be clear that we L2 throughout the paper.
>
>  > Is the method able to solve false object detections by Monet?
>
> We have used MONet quite extensively but are not familiar with the problem of  “false detections”, is this similar to over segmentation? If so, it is possible for the alignment and transition module to encourage less over segmentation since it is easier to model the dynamics of fewer objects (as you suggest above), however, we have not looked into this in detail.
>
>  > Can you clarify how the model is able to improve out-of-distribution generalization or few-shot transfer?
>
> When we talk about generalisation, we mean generalising to longer unroll lengths than those seen during training. For example, a model is typically trained with 4 encoding steps and to unroll for 6 steps and at evaluation time we demonstrate that the model can unroll for many more time-steps than those seen during training. On the robotics dataset we show unrolls of 100 time-steps.
>
> > Figures are difficult to see, frames are really small making very hard to appreciate the results.
>
> Yes, thank you for pointing this out. We are trying to show results over many time-steps (> 100 in the case of the robotics data) and object-slots. We have removed some intermediate time-steps to make the images more clear and we can remove additional time-steps to make the images more clear. We also include additional larger images in the Appendix. We can also remove some of the white spaces around the images to make them more clear.
>
> > There are problems with references, several of them do not include information about publication venue.
>
> Thank you for pointing this out, we will update these.
>
> > Missing reference: Tracking the world state with recurrent entity networks.
>
> We will update the paper to include this reference.

---

### Official Review · Reviewer_eFBe · 2021-07-19

**Rating:** 5
**Confidence:** 4

**Summary:**

The goal is to learn models that can parse object entities of a scene from third person views in an unsupervised manner. A key challenge in prior work is to enforce consistent parsing over time. To this end, this paper proposes an explicit alignment module, together with a transition module on top of it. By training the whole model end-to-end, the model is claimed to able to extract consistent object-level features while aligning the entities of these features over time. Evaluation is conducted on one simulation based dataset and one real world dataset. The proposed approach is shown to outperform a prior method [31] on perceptual and image segmentation based metrics.

**Limitations And Societal Impact:**

No discussions on potential negative societal impact.

**Main Review:**

### Formatting Issue
Before proceeding with any reviewing, I want to point out that this submission has a formatting issue. The main PDF contains appendix content after the references and paper checklist (see page 12 to 21). This extra content is only allowed to show up in supplementary, but is somehow presented in the main PDF here.

Even with this flaw, I'll still perform my reviewer duty and provide the review below.

### Comments
- [Fig. 1] Better explain what these images represent in the caption. These are images visualized by MONet's decoder? So the actual task is not video frame prediction (which is what it looks like in the first glance), but rather to segment the scene based on objectness (which again has non-trivial connection to these images).
- [Fig. 1] For OP3, is a MONet decoder additionally trained to decode it's representation? Otherwise how could a pre-trained MONet decoder work out of the box?
- [L120] "During the encoding phase the transition model is fed aligned, observed object representations, $z^a_t$, and actions, $a_t$, to predict ..." -> What is this "action"? Where does it come from? Is this an "action" of a policy in the sense of RL? The proposed approach is just doing unsupervised learning and thus should not have any RL stuff right?
- [L129] What do you do with the second $\Delta^m_t$ (predicted in the unroll step)? Is it also used to update $m_t$ following $m_{t+1}=m_t+\Delta^{m}_t$ (L128) after the same update is done for the first $\Delta^m_t$ from the encoding step?
- [L146-148] "This is a spatial mixture loss parameterised by $\sigma_{bg}$ and $\sigma_{fg}$, ... that go into the mixture loss." -> This is unclear. The authors should clearly explain the high level mechanism of the MONet loss without needing the readers to go to [2].
- [L154-161] "Alignment module losses ... for all experiments presented in this paper." -> This is also unclear. Why can minimizing the difference between $z^a$ and $z^d$ encourage alignment (or i.e. temporal consistency)? What is the intuition behind that?
- [L228-232] Since the main baseline is [31], why not just compare on the dataset used in [31] (and potentially just use the numbers in their paper)? Why is this 3D Playroom environment dataset needed? With that you also need to re-train [31] on this new dataset right?
- [L228] "3D Playroom environment [14,15]" -> Wrong citation. Should be changed to the following:
  - Josh Abramson, Arun Ahuja, Iain Barr, Arthur Brussee, Federico Carnevale, Mary Cassin, Rachita Chhaparia, Stephen Clark, Bogdan Damoc, Andrew Dudzik, Petko Georgiev, Aurelia Guy, Tim Harley, Felix Hill, Alden Hung, Zachary Kenton, Jessica Landon, Timothy Lillicrap, Kory Mathewson, Soňa Mokrá, Alistair Muldal, Adam Santoro, Nikolay Savinov, Vikrant Varma, Greg Wayne, Duncan Williams, Nathaniel Wong, Chen Yan, Rui Zhu. Imitating Interactive Intelligence. arXiv preprint arxiv:2012.05672, 2020.
- Overall the image quality in the paper is not at the publication level.
  - Fig. 1 -> Images too blurry.
  - Fig. 2 -> The text between "Transition module" blocks are illegible.
  - Fig. 4 -> Images too small.
  - Fig. 7 -> Images too blurry. Hard to tell what the content is.
- Please add the venue of the following citations:
  - [31] -> CoRL 2019.
  - [33] -> arXiv preprint.
  - [34] -> arXiv preprint.

### Justification of Rating
On the up side, the paper is addressing an important and challenging direction. On the downside, the quality of the presentation can be improved (see the unclear details above). Also the intuition behind the design of the loss functions is unclear, and the use of a new dataset for experimental evaluation is unjustified. Given these concerns, the initial rating is kept at slightly below the borderline.


**Time Spent Reviewing:**

5

---

> ### Author Response · Authors · 2021-08-06
> **Response to Reviewer eFBe**
>
> Thank you for taking the time to review our paper and for providing constructive feedback. We have addressed your points below:
>
> > The goal is to learn models that can parse object entities of a scene from third person views in an unsupervised manner.
>
> This only captures part of what our model does. We present an object-centric transition model that has three components: (1) a scene decomposition module, (2) an alignment module and (3) an object-based transition module. What we present is much more than just a segmentation model (as suggested in the reviewer’s first point below).
>
> > [Fig. 1] Better explain what these images represent in the caption. These are images visualized by MONet's decoder? So the actual task is not video frame prediction (which is what it looks like in the first glance), but rather to segment the scene based on objectness (which again has non-trivial connection to these images).
>
> It seems that the reviewer thinks that our model only performs segmentation, but this is only part of what our system does and not even the main contribution. We present an object-based transition model that makes n-step object representation predictions, we then visualise the predicted object representations using MONet’s decoder (as described in Section 4.1, line 250, Figure 3, 4 and 5). We will update Section 4.1 to make it more clear that we use this visualisation in all parts of the paper.
>
> > [Fig. 1] For OP3, is a MONet decoder additionally trained to decode it's representation? Otherwise how could a pre-trained MONet decoder work out of the box?
>
> OP3 has its own decoder. We have followed the specification in the original OP3 paper.
>
> > [L120] "During the encoding phase the transition model is fed aligned, observed object representations, zat, and actions, at, to predict ..." -\> What is this "action"? Where does it come from? Is this an "action" of a policy in the sense of RL? The proposed approach is just doing unsupervised learning and thus should not have any RL stuff right?
>
> We collect a dataset using a pre-trained agent, there is no RL loss used in our model. The details are given in Section 4.1, lines 236-238: “A dataset of observation-action trajectories, $(x_t, a_t)_{t=0,1,...,20}$, is generated by an agent taking actions according to a learned policy in a procedurally generated room.”
>
> > [L129] What do you do with the second $\Delta_{tm}$ (predicted in the unroll step)? Is it also used to update $m_t$ following $m_{t+1}=m_t+ \Delta_{tm}$ (L128) after the same update is done for the first $\Delta_tm$ from the encoding step?
>
> There is only one $\Delta_{tm}$. Encoding and unrolling are done on different time-steps. For example, you encode for $t=0:k$ and then unroll for $t = k+1:T$. In our experiments (as stated in the paper) we encode the first 4 time-steps and then unroll for the following 6 time-steps (during training). During the unroll, $m_t$ is updated in the same way as in the encoding (as the equations show). We will update the paper to make this more clear.
>
> > [L146-148] "This is a spatial mixture loss parameterised by $\sigma_{bg}$ and $\sigma_{fg}$, ... that go into the mixture loss." -> This is unclear. The authors should clearly explain the high level mechanism of the MONet loss without needing the readers to go to [2].
>
> Thanks for this suggestion, we will add more detail about the MONet loss.
>
> > [L154-161] "Alignment module losses ... for all experiments presented in this paper." -> This is also unclear. Why can minimizing the difference between za and zd encourage alignment (or i.e. temporal consistency)? What is the intuition behind that?
>
> Thank you for this question. We will add additional information to the paper. From a mathematical perspective:\
> You can either compute the object representations at the next time-step via predicting the dynamics:\
> 		 $$z^d_{t+1} = z^a_t + \Delta_t$$ \
> Or by predicting how the objects at the next time-step should be aligned: \
>           		$$z^a_{t+1} = A_t z_t$$
>
> where $z^a_t$ are the aligned objects at the current time step, $\Delta_t$ is the dynamics update, $z_t$ are the unaligned objects and $A_t$ is a permutation matrix.
>
> However, the object representations and their ordering at any time-step should be the same, independent of how you compute them. Therefore we want to find an $A_t$ and a $\Delta$ for which $z^d_{t+1}$ is the same as $z^a_{t+1}$, and hence we train the model to minimise the difference between them.
>
> > [L228-232] Since the main baseline is [31], why not just compare on the dataset used in [31] (and potentially just use the numbers in their paper)? Why is this 3D Playroom environment dataset needed? With that you also need to re-train [31] on this new dataset right?
>
> We are primarily interested in developing transition models that could be used by embodied agents in a 3D world. This means having an agent moving around a 3D environment and having to cope with objects moving with respect to the agent's filed of view and partial observability. Additionally the dataset in [31] only provides start and end states for each object when an action is taken, but not the intermediate steps, which is unlike the real world where we see more smooth trajectories. Yes, we did re-train OP3 on our dataset and we put the details in the Appendix.
>
> > [L228] "3D Playroom environment [14,15]" -> Wrong citation. Should be changed to the following:
> Josh Abramson, Arun Ahuja, Iain Barr, Arthur Brussee, Federico Carnevale, Mary Cassin, Rachita Chhaparia, Stephen Clark, Bogdan Damoc, Andrew Dudzik, Petko Georgiev, Aurelia Guy, Tim Harley, Felix Hill, Alden Hung, Zachary Kenton, Jessica Landon, Timothy Lillicrap, Kory Mathewson, Soňa Mokrá, Alistair Muldal, Adam Santoro, Nikolay Savinov, Vikrant Varma, Greg Wayne, Duncan Williams, Nathaniel Wong, Chen Yan, Rui Zhu. Imitating Interactive Intelligence. arXiv preprint arxiv:2012.05672, 2020.
>
> Thank you, we will update this citation.
>
> > Overall the image quality in the paper is not at the publication level. \
> Fig. 1 -> Images too blurry. \
> Fig. 2 -> The text between "Transition module" blocks are illegible. \
> Fig. 4 -> Images too small. \
> Fig. 7 -> Images too blurry. Hard to tell what the content is.
>
> We apologize to the reviewer for the small images, and for any strain this may have caused. It is part of the nature of this work that we generate large numbers of images, and it is very hard to present these within the page limits. The images should be high resolution, however, and clear if viewed on screen and zoomed in.
>
> - Fig 1: The baselines OP3 and OT results are blurry because of the nature of the models. Our results, OAT, are much more clear and this is a result of the alignment and the object-level losses that we propose.
> - Fig 2: Thank you for pointing this out, the text says $L_{\text{Transition model}}$. We will update this figure to make the loss more clear.
> - Fig 4: We show larger versions of these in Figure 11 of the Appendix. However, we will remove the white spaces between images to make the figure more clear. Thank you.
> - Fig 7: We have removed intermediate frames to make the unroll as clear as possible. We are showing unrolls for 100 time-steps in this figure. We can remove additional intermediate time-steps to make the figure more clear.
>
> > Please add the venue of the following citations:
> [31] -> CoRL 2019. \
> [33] -> arXiv preprint. \
> [34] -> arXiv preprint.
>
> Thank you, we will do this.

---

### Decision · Program_Chairs · 2021-09-28

**Decision:**

Accept (Poster)

**Comment:**

Reviewers agreed that this is a solid paper that deserves acceptance. Authors are highly encouraged to address the key comments reported by reviewers as well as to implement all the improvements (as indicated by authors in the rebuttal) in the final camera-ready version.

**Consistency Experiment:**

NeurIPS has a long history of experimentation. In 2014, NeurIPS ran an experiment in which 10% of submissions were reviewed by two independent committees to quantify the randomness in the review process. This year, we repeated a variant of this experiment to see how the quality of the review process has changed over time.  This paper was part of the experiment and was therefore assigned to two committees (consisting of reviewers, an Area Chair, and a Senior Area Chair) that reached independent decisions.  If both committees made the same recommendation, this recommendation was followed. If a single committee recommended acceptance, the paper was accepted (with the exception of a few cases in which the other committee identified what we considered a fatal flaw, e.g., an error in a key result).

This copy’s committee reached the following decision: **Accept (Poster)**

The other committee assigned to the paper recommended **Reject**.  You can find the other set of reviews, along with any follow up discussion with the authors here:
https://openreview.net/forum?id=X17EOUP2Cgt